# A Qualitative Study of Black College Women's Experiences of Misogynoir and Anti-Racism with High School Educators

**Seanna Leath [1,*], Noelle Ware [1], Miray D. Seward [2], Whitney N. McCoy [2], Paris Ball [2] and Theresa A. Pfister [2]**

[1] Psychology Department, University of Virginia, Charlottesville, VA 22904, USA; nw5ty@virginia.edu
[2] School of Education and Human Development, University of Virginia, Charlottesville, VA 22904, USA; ms2ma@virginia.edu (M.D.S.); wnm3mx@virginia.edu (W.N.M.); pb9zj@virginia.edu (P.B.); tap5g@virginia.edu (T.A.P.)
* Correspondence: sl4xz@virginia.edu

**Abstract:** A growing body of literature highlights how teachers and administrators influence Black girls' academic and social experiences in school. Yet, less of this work explores how Black undergraduate women understand their earlier school experiences, particularly in relation to whether teachers advocated for their educational success or participated in discriminatory practices that hindered their potential. Using consensual qualitative research (CQR) methods, the present semi-structured interview study explored the narratives of 50 Black undergraduate women (mean age = 20 years) who reflected on their experiences with teachers and school administrators during high school. Five discriminatory themes emerged, including body and tone policing, exceptionalism, tokenization, cultural erasure in the curriculum, and gatekeeping grades and opportunities. Three anti-racist themes emerged, including communicating high expectations and recognizing potential, challenging discrimination in the moment, and instilling racial and cultural pride. Our findings highlight the higher prevalence of discriminatory events compared to anti-racist teacher practices, as well as how the women's high school experiences occurred at the intersection of race and gender. The Authors discuss the need to incorporate gender and sexism into discussions of anti-racist teacher practices to address Black girls' experiences of misogynoir. We hope our findings contribute to educational initiatives that transform the learning landscape for Black girls by demonstrating how educators can eliminate pedagogical practices that harm their development.

**Keywords:** black women; high school; teachers; misogynoir; anti-racism; consensual qualitative research

High school educators are significant gatekeepers for youth's future educational and occupational success, setting the path for students' entry into college or the workforce (Roderick et al. 2011). Yet, race, gender, and socioeconomic disparities in educational access undermine college attainment for many high school students from historically marginalized groups in the U.S. (Baker et al. 2018). For instance, while the overall college enrollment rate for 18- to 24-year-olds increased from 35% in 2000 to 41% in 2018, the college enrollment rate was significantly higher for students who were Asian (59%) and White (42%) compared to Black (37%) and Hispanic (36%) (Digest of Education Statistics 2019). Furthermore, evidence suggests that Black girls receive inadequate support—academically, socially, and psychologically—in many high schools across the country (Carter Andrews et al. 2019; Morris 2016a; Neal-Jackson 2018; Watson 2016). Teachers and school administrators play a critical role in disrupting inequitable school processes for Black girls (Morris 2019); in particular, by supporting students' academic and social skills in ways that grant them access to educational benefits and opportunities (Neal-Jackson 2018). For some young Black women, more than others, high school educators play a critical role in regards to advanced degree attainment, career choice, and lifetime earnings (Patton and

Croom 2017). Yet, less research attends to how Black girls navigate high school and achieve their educational goals despite school-based marginalization (for notable exceptions, see Anderson 2020; Carter Andrews et al. 2019). To build upon extant literature, we explored the narrative reflections of Black undergraduate women regarding their experiences of misogynoir and anti-racism with high school teachers.

Few researchers have considered how young Black women process their experiences with teachers after leaving their classrooms, or the extent to which it informs their motivation and sense of college readiness (e.g., Anderson and Martin 2018). There is a need for critical scholarship on the academic experiences of high-achieving Black female learners to promote more nuanced representations of Black women's achievement and Black girls' educational trajectories (Anderson 2020; Evans-Winters 2014; Young 2020). In the present qualitative study, we sought to explore how a sample of Black undergraduate women in the U.S. made sense of their high school journey, particularly regarding the positive and negative experiences that participants described with teachers and school administrators. This reflective approach allowed us to ask the women questions about how teachers helped or hindered them throughout high school, focusing especially on Black women's discussions of misogynoir (i.e., the ways in which racism and sexism intersect to produce racialized gendered violence and harm against Black women and girls; Trudy 2014) and anti-racist teacher practices (i.e., efforts that go beyond education reform, to transform the structural inequities that maintain racism; Lynch et al. 2017) among secondary school educators. We draw on critical race feminism (CRF) as a theoretical tool to analyze Black women's racialized and gendered experiences with their high school teachers (Evans-Winters and Esposito 2010).

## 1. Critical Race Feminism and Black Girls' Education

Given our focus on combatting Black women and girls' experiences of race and gender oppression in educational settings, we incorporated Evans-Winters and Esposito's (2010) framework of critical race feminism (CRF). CRF is a branch of critical race theory (CRT) scholarship, which (1) highlights how race and racism are endemic to U.S. society; (2) challenges dominant ideologies of racial neutrality and meritocracy; (3) involves a clear commitment to activism and social justice; and (4) centers on the experiences and voices of structurally marginalized peoples and communities (e.g., Solórzano and Yosso 2002). Despite its utility in framing how to improve the U.S. educational landscape by addressing racism (i.e., Delgado and Stefancic 2017; Ladson-Billings 2016), female scholars of color have described the conceptual limitations of CRT for addressing the intersections of race, class, and gender (Lindsay-Dennis 2015; Wing 1997). As Evans-Winters and Esposito (2010) noted, "because feminist epistemologies tend to be concerned with the education of White girls and women, and raced-based epistemologies tend to be consumed with the educational barriers affecting Black boys, the educational needs of Black girls fall through the cracks" (p. 12).

The five main tenets of CRF most relevant to the current study include, (1) the belief that Black women and girls' experiences are different from the experiences of men of color and those of White women; (2) the focus on Black women and girls' experiences of discrimination at the intersection of race, class, and gender; (3) the assertion that multiple identities and ways of knowing inform Black women and girls' experiences; (4) the necessity of multidisciplinarity in writing about Black women and girls; and (5) a call for theory and practice that studies gender and racial oppression in an effort to combat its effects. CRF provides a framework to analyze Black women and girls' misogynoiristic experiences in school settings and lays the groundwork to consider how to actively promote educational change at the micro- and macro-level by integrating gender and social class into anti-racist discourse. We drew on CRF to examine, for instance, how Black women and girls' bodies and language are surveilled, policed, and controlled in schools (Anderson 2020; Haynes et al. 2016; Wun 2014), as well as the extent to which Black women and girls contest such experiences through acts of resistance and resilience (Evans-Winters 2005,

2014; Ford et al. 2019; Greene 2016). Finally, we hope that our findings will not only contribute positively to the empirical literature on women and girls of African descent, but also be instructive for scholars and educators who want to transform pedagogical practices in the classroom and broader school communities.

## 2. The Impact of Teacher Misogynoir on Black Girls' Schooling Experiences

Over the past few decades, awareness of the urgent need to address racial bias and discrimination in schools has grown, resulting in a number of anti-racist school initiatives and educational interventions (e.g., De Lissovoy and Brown 2013; Gillborn 2005; Ladson-Billings 2016). Yet, a growing body of evidence demonstrates that many of Black girls' school-related challenges with teachers and school leaders involve discriminatory practices at the intersection of racism and sexism, such as misogynoir (Carter Andrews et al. 2019; Davis 2020; Morris 2016a; Neal-Jackson 2018; Watson 2016). Misogynoir refers to the ways in which racism and sexism intersect and contribute to harm against Black women and girls through interpersonal encounters and institutional structures (Trudy 2014). Understanding the functionality of misogynoir as an ideological construct can help scholars and educators contextualize the innumerous ways that Black women and girls are subject to inequitable and harmful treatment in educational settings and broader society (Bailey and Trudy 2018). For instance, educators have lower academic and behavioral expectations for Black women and girls that can erode the development of healthy, caring relationships and a strong sense of connection to the learning environment (Archer-Banks and Behar-Horenstein 2012; Blaisdell 2020; Patton and Croom 2017). Carter Andrews et al. (2019) found that Black girls (9th–12th) expressed the impossibility of meeting academic and behavioral expectations often predicated on notions of white femininity; the expectation of being "perfect and white" was a feat that the girls in the study could never attain, which left them feeling underappreciated and rejected, with less moral and material support from the educators in their school.

In addition, misogynoiristic ideologies undergird how and why some teachers police Black girls' style of dress (i.e., receiving dress code violations for clothing choices; Morris 2016a) and hairstyles (i.e., being characterized as nappy or unacceptable; Jacobs and Levin 2018) in ways that are distinct from boys and girls from other racial groups; researchers demonstrate the emotional and psychological harm extolled on Black girls for simply existing in their bodies (Carter Andrews et al. 2019; Epstein et al. 2017). For instance, Watson (2016) interviewed six Black girls who were in their senior year at a predominantly Black and Hispanic high school and found that many of the girls believed that teachers were more concerned with their adherence to school rules than their safety and wellbeing. As one adolescent girl explained, "You could be doing what you have to do, walking through the hallway and they just put you in the safe room (a room where truant and difficult students are held)." Her experience was not unique, but instead, reflected how school authority figures emphasized controlling Black girls' social behaviors rather than supporting their academic achievement (e.g., Froyum 2010). Indeed, a number of studies indicate that Black girls in the U.S. experience harsher punishment and disciplinary consequences in school compared to girls from other racial groups (i.e., Black girls are six times more likely to be suspended than White girls; Blake et al. 2011; Hines-Datiri and Carter Andrews 2020; Wun 2014).

Misogynoir also contributes to the adultification of Black girls (i.e., belief that Black girls are more independent compared to White girls and need less nurturance, protection, and support; Carter et al. 2018; Epstein et al. 2017), which can increase their risk for discriminatory experiences in school settings (e.g., Seaton and Carter 2019). In a report from the National Women's Law Center (NWLC 2018) on dress code bias in D.C. schools, the authors found that adults held negative stereotypical beliefs that Black girls were more sexually provocative because of their race, and thus more deserving of punishment for a low-cut shirt or short skirt. Other scholars have found that Black girls who were more physically developed or curvier than their peers were viewed as more promiscuous,

which has been related to lower academic and behavioral expectations from teachers (Carter et al. 2018). As Jackson (2020) stated, "Black girls have been told all their lives that they are "fast" even when they are simply existing, as if they are responsible for the way their bodies develop" (p. 1). In all, this work highlights how Black girls' behaviors are misconstrued and over-policed by educators, which can open the door to pushing Black girls out of school (i.e., disengagement from the classroom and suspension or expulsion; Morris 2016a). When teachers are unaware of the ways that misogynoiristic ideologies influence how they treat Black female students, they are unable to recognize girls' agentic behaviors as positive traits rather than non-compliance (Annamma et al. 2019). Researchers suggest that teachers must play a key role in transforming school contexts into supportive spaces by dismantling the types of educational practices that harm Black girls' learning and development (Annamma 2015; Evans-Winters and Esposito 2010).

### 3. Moving towards Anti-Racist Educational Praxis

Researchers suggest that many Black girls and their families do their best to assimilate and navigate the cultural and hegemonic demands of most schools (Holland 2012; Lewis and Diamond 2015). Yet, it is school leaders and educators' responsibility to address how racism influences day-to-day routines in schools and adopt anti-racist strategies that maximize the educational opportunities available to students (Neal-Jackson 2018). Below, we review three key domains of anti-racist educational praxis that—when integrated—should catalyze more just and equitable schooling environments for Black girls. First, we review literature on cultivating Black girls' "oppositional gaze" (Hooks 1992; Jacobs 2016) to help them interpret the misogynoiristic experiences they have in school within a broader understanding of the historical underpinnings of racism and sexism. Second, we highlight the role that teachers can play by learning more about Black girls' lived realities and implementing anti-racist pedagogical practices (Greene 2020; Lane 2017). Third, we describe how equity-oriented school leadership is critical for disrupting school structures that marginalize and impede the academic success of Black girls.

### 4. Developing Black Girls' Critical Ways of Knowing

Teachers can craft educational pathways of agency for Black girls by developing their critical ways of knowing. Specifically, teachers can help their Black female students form nuanced understandings of how social institutions perpetuate narratives of deficiency about Black women and girls (Edwards et al. 2016; Evans-Winters and Esposito,2010; Morris 2016a). Oft referred to as an "oppositional gaze" in studies on media representation (Hooks 1992; Jacobs 2016), Black girls in high school are developmentally old enough to develop a critical consciousness about negative media portrayals of Black women (i.e., emotionally unstable, hypersexual, and unintelligent; Walton 2013) and form their own counter narratives that affirm their self-worth (Jacobs 2016; Lane 2017; McArthur and Lane 2019). Developing an oppositional gaze enables Black girls to recognize sexist messages and expectations from others—consider how their gendered experiences are racialized—and craft the internal confidence to define themselves on their own terms (Kelly 2018). This is critical, given that constant exposure to negative stereotypical messages about Black women and girls relates to cultural dissonance for some Black girls, who begin to see themselves as less valuable in society compared to girls from other racial/ethnic groups (Edwards and Esposito 2016; Kelly 2018; Price-Dennis et al. 2017). Thus, teachers who encourage Black girls to develop an "oppositional gaze" in their consumption of mainstream media and school curricula, may mitigate the oppressive influence of racism and sexism on Black girls' developing identity beliefs.

For example, Muhammad (2015) used writing as a platform to allow adolescent Black girls (12–17 years) to create self-representations that countered deficit-based stereotypes and described the types of social changes they would like to see in the world (e.g., improving social conditions so that Black girls would not be overtaken by self-doubt, p. 239). The students used the writing activities to explore power dynamics, relate to their own lived experiences, and make sense of their gender and racial identities. The participants wrote multiple, complex pieces, freely constructing understandings of their identities in intersectional ways that included ethnicity, gender, sexuality, and kinship and community representations. A growing number of similar scholars are beginning to interrogate and expand on the ways that culturally responsive teachers can provide Black girls with more opportunities to learn about and incorporate their histories, identities, and knowledge into instructional activities. Educators play a critical role in positioning Black girls as experts of their own experiences, which includes preparing them to understand and reject deficit-based stereotypes of hypersexuality, behavioral problems, and simultaneous hyper- and invisibility (Hope et al. 2015; Jacobs 2016; Walton 2013). In relation to the present study, we suggest that teachers who draw upon anti-racist pedagogical practices can establish classroom environments that allow Black girls to critically examine and affirm their status in society. Educational spaces can be sites of affirmation that promote the holistic and positive development of Black girls' identity beliefs (e.g., Brown 2009; McArthur and Lane 2019; Nyachae 2016).

## 5. Building a Politicized Ethic of Care among High School Teachers

In addition, a guiding component of anti-racist curricula is the humanization of Black girls in the classroom through a politicized ethic of care (i.e., ideological posture where teachers facilitate the social and intellectual empowerment of students through authentically caring and healing pedagogies; Lane 2017). Acknowledging how mainstream educational practices are harmful to Black girls and actively working to change those practices is inherently anti-racist in nature. Prior research has shown that Black girls understand when they are being policed in the classroom through negative social interactions with their teachers (e.g., Froyum 2010; Morris 2007). A growing body of work highlights how Black girls benefit from, and respond positively to, qualified educators who exercise patience, kindness, and understanding in the classroom (Joseph et al. 2017; Joseph et al. 2019; Young 2020). For instance, Joseph et al. (2019) found that Black, adolescent girls (12–17 years) greatly valued collaborative work in mathematics classrooms, as it offered girls the opportunity to connect to one another and share their knowledge about course content. In addition, mathematics teachers who allowed Black girls to make mistakes without punishment and participated in one-on-one instruction with struggling students allowed the students to learn challenging concepts without being shamed in the classroom.

Scholars suggest that moving away from mainstream educational praxis that marginalize and punish Black girls in the classroom will provide more room for academic growth and development (Morris 2019; Price-Dennis et al. 2017). Morris (2016a) found that positive social interactions and collaborative lesson plans with teachers helped Black girls in grades K–12 feel more welcome and safe in school; when Black girls felt safer in the school environment, they also felt a stronger sense of belonging and more academic motivation. In other studies, teachers who asked for student feedback, displayed interest in the lived experiences of their students, and allowed students to steer the direction of their own learning were seen as more accepting (Joseph et al. 2019; Price-Dennis et al. 2017). Thus, educators who combine positive, humanizing interactions and acknowledge the barriers that Black girls face in education can foster positive identity development processes, which may in turn, increase academic success (Joseph et al. 2019; Young 2020). Black girls deserve teacher–student relationships and learning environments that promote critical thinking, collaboration, and positive socioemotional development (Grey and Harrison 2020; Watson 2016).

## 6. Dismantling School Structures That Harm Black Girls

While equipping them with knowledge of social structures and building positive teacher–student relationships are integral components in Black girls' academic success and wellbeing (Hope et al. 2015; Price-Dennis et al. 2017), moving towards anti-racist educational praxis requires that we eliminate schooling policies and practices that harm Black girls. In addition to facilitating positive interpersonal processes in the classroom, researchers also highlight how teachers' investment in anti-racist and culturally responsive pedagogy support Black students' engagement (Byrd 2016; Irby et al. 2019). Anti-racist curriculum decenter Eurocentric norms on knowledge and pedagogy (i.e., prioritizing White authors, ignoring cultural context, and grading students harshly for misunderstanding material) by replacing it with literature from a variety of cultures with different values and norms (Price-Dennis et al. 2017). Researchers suggest that teachers must "consider whose perspectives are at the core of their curriculum, who put them there, and why—what are the politics within their subject?" (Schwartz 2020, p. 1). These scholars intimate that moving towards anti-racism in schools requires that teachers illuminate the long-standing history of U.S. classrooms as sites of oppression for Black girls and other students of color (McArthur 2018, 2019).

In addition to pedagogical choices in the classroom, scholars assert that teachers must collaborate with other school leaders to demonstrate their commitment to creating equitable and racially just schools (Irby et al. 2019; Ladson-Billings and Tate 1995; Schwartz 2020). While many schools have published "diversity and inclusion statements," anti-racist schooling contexts require that educators do the difficult work of actively rooting out policies that disproportionately affect Black students (e.g., academic tracking, disciplinary sanctions, attendance policies, and dress codes; Soave 2021). While some schools have incorporated trainings on cultural responsiveness or professional development courses on implicit bias (Flynn et al. 2018), evidence suggests that these efforts are usually voluntary for teachers and largely dependent on the extent to which teachers themselves consider the trainings valuable (Lensmire et al. 2013). In addition, these school-wide efforts are too limited; a recent national survey from Education Week found that 82% of K–12 educators reported that they had not received anti-racist professional development and 59% said that they had neither the training nor the resources necessary to support the implementation of an anti-racist curriculum (Schwartz 2020).

## 7. The Current Study

Extant literature addresses the historical and social factors that influence Black girls' educational experiences during K–12 (Grey and Harrison 2020; Neal-Jackson 2018). Findings indicate that teachers' racial and gender bias and discrimination exert a significant influence on Black girls' achievement processes and psychosocial wellbeing (Archer-Banks and Behar-Horenstein 2012; Brown 2011; Hines-Datiri and Carter Andrews 2020); conversely, teachers who invest in Black girls' potential and work to address racism in schools, can support their academic growth and socioemotional wellbeing (Anderson 2020; Greene 2020). However, less is known about how young Black women interpret and process the experiences they had with teachers in their earlier years of schooling. Thus, the purpose of the present qualitative study was to draw upon critical race feminism as a theoretical lens to explore how Black undergraduate women described their academic and social experiences with high school educators, with particular attention to misogynoir and anti-racism among teachers. We hope the insight from the young women in our study will help scholars and practitioners who want to alter and expand our conceptualization of Black girl freedom in education.

## 8. Method

### 8.1. Participants

The sample included 50 Black women (ages 18–24 years, M = 20 years, SD = 1.5 years) enrolled at one of two predominantly White institutions in the U.S. Twenty-one women attended a public institution in the Midwest (5% Black/African American) and the remaining 29 attended a selective, public institution in the Southeast (6% Black/African American). At the time of data collection, seven women were in their first year of college, 17 were in their second year of college, 15 were in their third year college, and 11 were in their fourth year of college. Ethnically, 23 identified as African American, 22 identified as African (i.e., women from Nigeria, Ghana, Sudan, Ethiopia, Liberia, Sierra-Leone, Eritrea, and Cameroon), and three identified as biracial/multiracial. All the African women in our sample were naturalized citizens of the U.S., although a few mentioned that they had spent significant time in their parents' country of origin during their childhood. Self-reported social class status revealed that two women were from poor socioeconomic backgrounds, eight women were from working-class backgrounds, 35 were from middle-class backgrounds, and five were from upper class backgrounds (annual household income ranged from <$5000–$200,000, median = $65,000–$80,000). The women were in a range of disciplinary majors, including Education, Business, Math, Psychology, Black Studies, Sociology, Public Health, and Women and Gender Studies. See Table 1 for additional demographics.

**Table 1.** Participant demographic characteristics.

| Pseudonym | Class Year | Ethnicity | Social Class | Hometown | HS Racial Composition (% Black) |
|---|---|---|---|---|---|
| Afyia | 2nd | Eritrean | Middle Class | Suburban | 81%–100% |
| Akira * | 2nd | African American | Working Class | Urban | 61%–80% |
| Alexa * | 3rd | African American | Middle Class | Urban | <20% |
| Aliyah | 3rd | African American | Middle Class | Urban | 81%–100% |
| Alyssa * | 3rd | African American | Middle Class | Urban | 81%–100% |
| Amaya * | 2nd | African American | Middle Class | Urban | 61%–80% |
| Amber * | 2nd | Nigerian | Middle Class | Suburban | <20% |
| Angel | 4th | African American | Working Class | Small town | 81%–100% |
| Aniyah | 2nd | African American | Middle Class | Urban | 81%–100% |
| Brianna | 2nd | West Guinea | Working Class | Small town | <20% |
| Brie | 3rd | African American | Middle Class | Suburban | <20% |
| Brionna *,+ | 1st | Sudanese | Middle Class | Suburban | <20% |
| Candice | 2nd | Nigerian | Middle Class | Small town | <20% |
| Carey | 1st | Sierra-Leonean | Middle Class | Suburban | 41%–60% |
| Chloe * | 3rd | African American | Working Class | Urban | 81%–100% |
| Ciara | 3rd | Biracial | Upper Class | Suburban | <20% |
| Danielle * | 1st | Somali | Middle Class | Suburban | 41%–60% |
| Dashawna * | 1st | Cameroonian | Middle Class | Suburban | <20% |
| Desiree + | 4th | Liberian | Middle Class | Suburban | <20% |
| Destiny * | 3rd | Rwandan | Upper Class | Suburban | 41%–60% |
| Diamond | 3rd | Nigerian | Upper Class | Suburban | 21%–40% |

| | | | | | |
|---|---|---|---|---|---|
| Ebony [+] | 1st | Lebanon | Middle Class | Suburban | 41%–60% |
| Farah | 3rd | Ghanaian | Middle Class | Suburban | 81%–100% |
| Gabrielle * | 5th | African American | Middle Class | Urban | 81%–100% |
| Gemma | 2nd | Ethiopian | Middle Class | Suburban | 21%–40% |
| Grace | 2nd | Ghanaian | Middle Class | Rural | <20% |
| Hailey * | 1st | African American | Working Class | Urban | 81%–100% |
| Hannah * | 3rd | African American | Middle Class | Urban | <20% |
| Imani | 4th | African American | Middle Class | Urban | 81%–100% |
| Indigo * | 2nd | African American | Middle Class | Suburban | <20% |
| Isis | 4th | African American | Working Class | Urban | 81%–100% |
| Jada | 2nd | Biracial | Middle Class | Suburban | <20% |
| Jayla | 3rd | African American | Upper Class | Suburban | <20% |
| Jaleesa * | 4th | African American | Middle Class | Suburban | 21%–60% |
| Jordan [+] | 4th | African American | Upper Class | Suburban | 21%–60% |
| Kaja | 3rd | Ghanaian | Middle Class | Suburban | 81%–100% |
| Katrina * | 2nd | African American | Middle Class | Suburban | 61%–80% |
| Kayla | 4th | African American | Middle Class | Urban | 81%–100% |
| Kennedy | 4th | Biracial | Middle Class | Suburban | 41%–60% |
| Laila | 3rd | African American | Middle Class | Urban | 81%–100% |
| Lakeisha * | 3rd | Nigerian | Working Class | Suburban | 61%–80% |
| Madison | 3rd | Nigerian | Middle Class | Urban | 81%–100% |
| Makayla | 5th | African American | Middle Class | Urban | 81%–100% |
| Neveah | 2nd | Ethiopian | Middle Class | Suburban | 41%–60% |
| Noelle * | 2nd | Ethiopian | Middle Class | Suburban | <20% |
| Sydney | 2nd | Senegalese | Working Class | Suburban | 41%–60% |
| Tamika | 4th | Ghanaian | Middle Class | Suburban | 21%–40% |
| Taylor | 2nd | African American | Working Class | Suburban | 61%–80% |
| Tiana * | 2nd | African American | Working Class | Suburban | <20% |
| Trinity | 1st | African American | Middle Class | Small town | <20% |

Note: * Quoted in article. [+] Identified as bisexual—other women identified as heterosexual. HS Racial Composition = High school racial composition of the student body.

### 8.2. Procedures

After receiving university IRB approval from each location, the primary investigator (PI) sent weekly emails to Black student organizations until we scheduled the target number of participants for interviews. Students who identified as Black/African American women were eligible for participation in the study. At the first institution, the PI had research funds to interview approximately 20 women. Twenty-four women scheduled interviews (3 women canceled), and the 21 interviews were completed in the spring of 2019. At the second institution, the PI decided that theoretical saturation (i.e., point at which sampling more data did not lead to more information related to the main research questions; Saunders et al. 2018) had been reached after 29 interviews. Thirty-five women scheduled interviews (six women canceled), and the 29 interviews were completed in the fall of 2019. Black women who were interested in participating replied to the email to schedule a time to meet. The team conducted the individual interviews in safe, public

locations that allowed for private conversation (i.e., reserved conference room and PI office) and used iPads to audio record the women's narratives. The interview team at each institution consisted of the PI (a Black woman, S.L.), as well as an additional Black woman graduate student (M.W. and D.J.) per institution, who was trained in qualitative interview techniques. Interviews lasted between 45 and 90 min (M = 75 min). Before each interview, participants completed a survey with demographic items. The women were compensated $20 for study participation. The PI sent the audio files out for professional transcription, and the original interviewer reviewed the transcript to ensure accuracy.

### 8.3. Interview Protocol

We collected the narrative data using a semi-structured interview protocol (see Appendix A). The protocol included a set list of questions about race and gender socialization messages, as well as supplemental questions that interviewers could ask participants based on the direction and flow of the conversation (Cohen and Crabtree 2006). This approach allowed the research team to collect reliable, comparative data on the topics of interest, while also providing the opportunity to identify new understandings through additional probing questions.

Each interview consisted of two main sections—the first section focused on race and gender socialization messages and the second portion focused on their awareness of stereotypes about Black women. Participants responded to questions about the socialization they received during childhood and adolescence from peers, family members, and school contexts regarding their identities as Black women. The interview questions relevant for the current study included, "During K–12, what types of messages did you receive about being a Black girl from people at school? This includes messages from peers, teachers, or other school figures." Based on their responses, interviewers probed about significant experiences with questions like, "Did you have a favorite teacher? What did you like about them?" Some participants discussed teacher experiences in high school throughout the interview, so we reviewed the transcripts in their entirety.

### 8.4. The Research Team

We recognize that researchers' identities and multiple roles influence the process and outcomes of research (Evans-Winters 2019), and so we offer insight into our social positionalities as scholars and our roles in the current project. The lead author and PI (S.L.), who was involved at every stage of the research process in the current project, is a Black, queer woman from a working-class background. She has conducted over seven years of research on the educational experiences of Black women and girls. The second author (N.W.), who was involved in the writing of the paper, is a Black, queer, non-binary doctorate student in psychology from a rural, working-class background. They have studied systemic structures and mental health with a focus on Black, disabled children for six years through the lens of educational psychology. The third author (MS), who was involved in the writing of the paper, is a Black woman from a middle-class background. She is a doctoral student in an educational psychology program and her research focuses on the positive development of Black women and girls, with a specific focus on student-athletes. The fourth author (W.M.), who was involved in writing the manuscript, is a Black woman from a middle-class background with a doctorate in educational psychology. Her research explores identity development for Black girls in educational settings, especially regarding informal science, technology, and math (STEM) education. The fifth author (P.B.), who was involved in the coding and writing of the paper, is a Black woman from a lower-income background. She is a doctoral student in a clinical and school psychology program and her research centers on the holistic development of Black women and girls. The sixth author (T.P.), who was involved in the coding and writing of the paper, is a white woman from a lower-income background. She has worked as a classroom teacher and college advisor, and she is currently working on her doctorate in educational psychology and developmental science.

*8.5. Coding Analysis Approach*

Consistent with recommendations from Hill (2012) on consensual qualitative research (CQR), the coding team (S.L., P.B., T.P., and N.W.) used a consensual coding approach supervised by the lead author. CQR methods rely on the depth and richness of constructivist qualitative methods and reaching consensus among the coding research team on core ideas from the data (e.g., Ponterotto 2005). First, the coding team (S.L., P.B., and T.P.) read through the transcripts and highlighted statements that addressed the research question: (1) In reflecting on their high school experiences, what types of interactions with high school teachers do Black women describe? The lead author reviewed all 50 transcripts, and P.B. and T.P. reviewed 25 transcripts each. After reviewing the transcripts, the coding team then met to review all highlighted statements from the 50 transcripts and reach a consensus on all statements.

A running log of inclusion and exclusion criteria (e.g., statements that referenced teacher experiences in middle school were excluded) was updated and available on a secure data server for reference. After finalizing the chunks of text, the PI (S.L.) uploaded the transcripts into Dedoose for coding. In accordance with suggestions by Hill (2012) on developing a codebook, each member of the chunking team (S.L., P.B., and T.P.) generated a list of domains that represented prominent themes from the data. We then met to compare our lists and define codebook domains. Upon reviewing our independent domain lists, we decided to organize the codebook by discriminatory teacher practices and anti-racist teacher practices. The final five discriminatory coding themes included: body and tone policing, exceptionalism, tokenization, cultural erasure in the curriculum, and gate-keeping grades and opportunities. The three anti-racist codes included: communicating high expectations and recognizing potential, challenging discrimination in the moment, and instilling racial and cultural pride. These domain codes were entered into Dedoose, a qualitative analysis program, and the PI (S.L.) and an additional trained graduate student (N.W.) coded half of the transcripts each. They met collaboratively and discussed any uncertainties during the coding process. Constant comparison of the assigned codes helped the pair refine and collapse categories, as well as generate new categories when necessary. This iterative analysis continued until we reached a consensus that the codes were accurate and concise reflections of the data.

Finally, the chunking and coding teams established the trustworthiness of the data in several ways. First, the coding process involved the consensus of multiple researchers, all of whom had thought about and discussed their potential biases throughout data analysis. The chunking team (S.L., P.B., and T.P.) met bi-weekly to discuss questions and disagreements in the coding process and develop a codebook. Then, S.L. and N.W. reviewed the domain codes and suggested several changes that were adopted, including the collapsing of categories that were poorly represented. After S.L. and N.W. completed coding, the original chunking team (S.L., P.B., and T.P.) reviewed the assigned codes to ensure that they captured the meaning of the individual responses. Finally, to engage in ongoing reflexivity, we participated in individual memoing and a series of dialogues about our biases and expectations before, during, and after the coding process.

## 9. Findings

How can teachers cultivate inclusive and anti-racist educational spaces to support Black girls' academic achievement and psychosocial growth? The women in our study described pervasive challenges with teacher and administrator misogynoir during high school that challenged their academic motivation and sense of belonging. Some women also recounted instances of anti-racist pedagogy and classroom practices among teachers. Below, we offer insight into some of the discriminatory teacher practices that sustain educational disparities among Black girls (Morris 2016a). We also enumerate specific ways that some teachers supported the Black girls in their classrooms and adopted the types of discursive practices that will be necessary to reimagine educational praxis. All names and

identifying information in this section, including teacher names, names of schools, and cities, are pseudonyms. To review the coding themes and excerpt examples, see Tables 2 and 3.

**Table 2.** Summary of discriminatory teacher practices.

| Discriminatory Teacher Practices | |
| --- | --- |
| **Theme** | **Excerpt Example** |
| **Policing physical appearance and language/tone** (*n* = 25, 50%) Refers to instances when teachers made comments or disciplined their physical appearance or behavioral mannerisms | "There's the normal stuff like cross your legs when you sit. Keep your legs closed. I've been told that I shouldn't be so loud. When it comes to dress code, make sure to cover yourself, wear stuff below the knees. Shoulders were sexy apparently, because nobody could have their shoulders out. Cover your bra straps. I feel like when it comes to being lady-like, they wanted me to water myself down." (Chloe, 3rd year, African American) |
| **Expecting girls to be exceptional** (*n* = 19, 38%) Refers to instances when teachers treated the women as though they were inherently different from other Black students | "I felt like a token throughout high school, and it would always give others the excuse to make these jokes. Am I here because y'all like me? Or am I here because y'all feel like I'm an exception? They would always say stuff that would make it like, "Oh, but you're one of the other Black people." And it's like, "What other Black people?"(Taylor, 2nd year, Nigerian) |
| **Tokenizing girls in the classroom** (*n* = 19, 38%) Refers to instances when teachers singled them out in class (i.e., expected them to speak for "Black people") | "To this day, I've noticed that if I'm unsure of an answer, I tend to not raise my hand and speak up compared to somebody else who might be unsure. Now I've realized that it's because I don't want to say something wrong. I was always the only Black person in the class and I feel like a representative, or they put that pressure on you." (Tiana, 2nd year, Ethiopian) |
| **Erasing racial diversity in curriculum** (*n* = 7, 14%) Refers to instances when the women perceived that Black history was neglected in course content | "It was mostly something that I was interested in reading about. I noticed myself doing research papers in high school about things that had to do with race, but I don't really think I was given that much in school." (Lakeisha, 3rd year, Nigerian) |
| **Gatekeeping grades and opportunities** (*n* = 7, 14%) Refers to instances when teachers gave out unfair grades or denied the women an opportunity without cause | "I asked my teacher, "Why isn't there any writings on my paper?" I wanted feedback and everybody else had feedback. She looked through my paper for two seconds and said, "Your paper is stapled twice." I was just like, "Is that really a 30 point deduction though?" She caused me to get my first C ever in a grade in a class. My parents called the school; the school was like, "Well, she's the teacher. We can't really do anything about it." (Danielle, 1st year, Somalian American) |

Table 3. Summary of anti-racist teacher practices.

| Anti-Racist Teacher Practices | |
| --- | --- |
| **Theme** | **Excerpt Example** |
| **Communicating high expectations and recognizing potential** (*n* = 13, 6%) Refers to instances when teachers provided encouragement around academic or personal goals and helped the women achieve those goals | "His name was Doctor E and he was an angel. He encouraged me to apply to Brown when I was applying to colleges. He told me, "No, apply to the Ivies—try. You don't have to stay here, you can do it." He really believed in me even though physics didn't come easy. He's one of the ones who made me think I really had a chance." (Indigo, 2nd year, African American) |
| **Challenging discrimination in the moment** (*n* = 4, 8%) Refers to instances when teachers called out students or other teachers/administrators for racial discrimination | "There was this one time that this White girl in my class got a question right and said, "Well Katrina can just have my bonus points." And my teacher was like, "You need them more than she does." Everyone just laughed. Like the other students in the class didn't always know that I was smart, but my teachers saw my grades so they knew." (Katrina, 2nd year, African American) |
| **Instilling racial and cultural pride** (*n* = 4, 8%) Refers to instances when teachers demonstrated a commitment to including racially diverse materials in the curriculum and helped the women begin to understand systems of power | "At first, I thought Black women were supposed to be on the sidelines of things, or if they weren't on the sidelines, then they were just going to be there but not heard. We had to read *Hidden Figures* for science class and I really like that book because it showed all these amazing things that had to happen for something that this country praises [walking on the moon]. But you didn't actually give any credit because it's—not only were they women—but they were Black women, so it's like they were just put in a lower class amongst our already secondary class." (Amaya, 2nd year, African American) |

## 10. Discriminatory Teacher Practices

*10.1. Policing Physical Appearance and Language or Tone (n = 25, 50%)*

The most common theme involved teachers and school administrators policing the women's physical appearance, body language, and tone of voice during interactions. Twenty-five women discussed how teachers admonished the ways that clothing fit their body, sent them to the principal's office for alleged dress code violations, and discouraged them from having an "attitude" in response to the unfair treatment. Taylor, a 2nd year Nigerian student, shared:

> I was on the debate team in high school, and wearing suits on the debate team was key. People judge you like, "Is she wearing a mismatched outfit from Forever 21? Or do her parents have money? Is that a tailored suit?" So I had on my best suit. I remember my debate teacher was like, "You look like a whore in that." The whole team heard it. Some people ignored it, but one of the girls who I was closer with was like, "Why would you say that? That's such a nice suit." My debate team coach was like, "Well, it's just so tight. You're showing your butt." First of all, he was like 50. Why is he looking at me? This is still one of those things where I'm like, "I can't believe I let that man talk to me like that."

As Taylor recounted the significance of physical appearance and clothing choices on her debate team, she highlighted the verbal assault and adultification that she received from her male debate teacher. The experience remained with her years later, including

most of her classmate's silent complicity in the exchange. In addition, the teacher–student power dynamic limited what Taylor might have been able to do without experiencing negative consequences. Other women in this theme detailed the harassment they experienced from teachers through subjective dress code violations. Amber, a 2nd year Nigerian student, said:

> I was the only Black girl in my school, which made it harder for me to accept my body because I didn't look like my friends when I was wearing exactly what they were wearing. I could be wearing the same skinny jeans as my friends, but teachers would have a problem with it or I would get called into the office. The idea of modesty was whipped on us. Like I was being immodest because I have this shape and I wore something that didn't hide it. I grew up feeling like my body was something to hide. In school, it would be a moral shame, like, "How dare you show your curves!"

Amber's narrative highlighted how body policing made the women feel hyperconscious during a time when they were already concerned with how they looked in comparison to peers. The negative attention from teachers, which sometimes resulted in disciplinary infractions, exacerbated their sense of isolation and exoticization in the school context. Such experiences were a significant challenge for participants, as the most frequent discriminatory teacher practice.

*10.2. Expecting Girls to Be Exceptional (n = 19, 40%)*

The second most frequent discriminatory practice involved nineteen women describing how teachers exceptionalized their academic behaviors in the classroom. Educators often suggested or stated explicitly that the girls "weren't like other Black students." In response, the women often received better treatment from these teachers compared to other Black students, but for some participants, it set the expectation that they maintain "perfection" to receive humanizing and supportive treatment from their teachers. In each example in this category, the women used the word "exception" to describe their positive experiences with teachers. Destiny, a 3rd year Rwandan woman, stated:

> I think teachers acknowledged me as an exception. As an exception, they were way more prone to remember my name, because I was Black and I was smart. They interacted with me differently than they would interact with other Black people at school. And I don't know how this was, but I guess teachers could tell by looking at me. They were like, "She's not African American." So our initial interactions were just different.

Destiny stated that teachers treated her better than other Black students because they "acknowledged her as an exception," which she mapped onto the anti-Black beliefs that some teachers possessed about African American students. She suggested that although teachers might not have known that she was Rwandan, specifically, they made a distinction between Black students who were African and those who were African American. While some women noted that they appreciated the positive teacher attention borne from this exceptionalism, others discussed the sense of pressure they felt from this type of treatment. Indigo, a 2nd year African American student, discussed the strong teacher support she received throughout K–12:

> It's been a lot of pressure. There's been a lot of pressure and I put a lot of pressure on myself because I strive for perfection always, which is an issue because I know you can't be perfect and I guess I don't want to disappoint and I feel like this is the one thing that I have control over and so I want to make sure I'm controlling it in all areas. I haven't dealt with it and I probably should, yeah. None of my friends have dealt with it really either, so we're all just kind of just like self-therapy, we just talk and…yeah, it's a problem.

Indigo's excerpt highlighted how many participants were surrounded by other high-achieving students who were striving for the best grades, strong recommendation letters for opportunities, and accolades within the college preparatory context. The exceptional treatment from teachers translated into an intense desire to succeed, in part, because they were concerned about letting others down if they did not maintain their academic excellence. High school represented a critical period for many of these students as they prepared for the next step in their journey towards occupational success. Yet, the exceptionalism in this category tokenized the young women and made them hyper aware of their Blackness. As with Indigo, the participants were unsure as to whether their teachers recognized their brilliance as students or considered them the best and brightest among the Black students whom they relegated to the margins.

*10.3. Tokenizing Girls in the Classroom (n = 19, 40%)*

Nineteen women discussed how teachers tokenized them as one of the only Black students, or Black girls, in the classroom. While related to exceptionalism, the tokenization theme focused on their numerical underrepresentation, rather than their academic strengths, in comparison to other Black students. Jaleesa, a 4th year African American woman, recalled:

> My history classes in high school were difficult because I would be one of maybe three to five Black students in classes of like 30 students. Whenever they would talk about slavery or the Civil War, students would always come to one of the Black students and ask us about it. Like we're the entire Black race! One of my teachers was like, "I'd like to hear from the Africans' point of view." I was like, "I am African American, and I don't know how to speak for my entire race to this class of 30 students when I don't even know what slavery was like. I'm learning about it just like everyone else." It was annoying to be feel like I needed to say what's right and what's wrong and what's racist and what's not. But it was also like, "The teacher's asking me a question, so what am I supposed to do?"

As in Jaleesa's example, the women discussed how this tokenization lessened their desire to engage in the classroom, often out of fear of saying the wrong answer or confirming negative stereotypes about Black intellectual inferiority. Her excerpt underscored the teacher's homogenization of ethnic diversity among Black people (i.e., "hear from the Africans"), and how the inherent power dynamic in teacher–student relationships challenged the women's ability to advocate effectively for themselves. While she had the language to name the teacher's discriminatory actions as an undergraduate, she did not have the same facility as a teenager in high school. This affected the women's experiences during the college admissions process, as well. Several participants discussed how teachers and counselors suggested that they would be accepted to college simply because they belonged to two historically underrepresented groups in higher education. Hailey, a 1st year African American student, shared:

> I still think about my teacher who told me, "You're a Black girl from the city. They don't have a lot of Black girls from the city, so they need you for their diversity thing." I try not to think about it as much—try not to let it get to me—but it can be very overwhelming.

This affirmative action rhetoric is documented in literature on interracial interactions among peers (e.g., Linley 2018), and the women highlighted how teachers communicated messages that reduced their college admissions success to institutional diversity quotas. At the institution that Hailey attended, affirmative action allowances or race-based admissions decisions had been overturned since the early 2000s, further demonstrating the illogical underpinnings of her teacher's assertion. While she might have been trying to assure Hailey of her college acceptance, the comment left her questioning whether she truly belonged at the prestigious state university.

*10.4. Lacking Racial Diversity in Curriculum (n = 7, 15%)*

While most of the discriminatory teacher practices involved direct interpersonal interactions between the women and their teachers, seven participants talked about the non-existent racial and cultural diversity in the school curriculum. A few of the women noted that they learned about prominent, mainstream historical figures such as Rosa Parks and Harriet Tubman, which still felt insufficient compared to the expansive Eurocentric history that they received on other topics. Noelle, a 2nd year Ethiopian woman, said:

> They shied away from the topic. The only time I'd say...we had an African Day. We would have different cultural things for that event. But it wasn't really talked about. In AP U.S. history, there would be a small part...my teacher was emphasizing it more than they [the school] would have him. But it was still a pretty small part. What is it called? The Civil Rights movement...he emphasized that a little bit more than past history teachers.

Noelle suggested that her advanced placement history teacher provided more information on the Civil Rights Movement than the school expected, highlighting how a Eurocentric (i.e., focusing on European culture or history to the exclusion of a wider view of the world; implicitly regarding European culture as preeminent; Charles 2019) curriculum fit within school standards. Four of the women also discussed how the absence of racial diversity in the curriculum translated to personal expectations regarding their language and behaviors in the classroom. For example, they believed that it was important to learn and master "Standard English," and did not realize the significant gaps in their teacher's class material. Alyssa, a 3rd year African American woman from a middle-class background, said:

> I would just say—it was always—as a Black girl in the real world, my teachers told me to make sure that I know how to speak properly. Sometimes I would get frustrated because these big words that are mainly for people who aren't from the inner city's vocabulary—aren't in mine. It just doesn't naturally come to me and I feel like I can't speak like them or I'm not on their level and it's gonna keep me back because I can't speak in that form.

Alyssa transferred from a predominantly Black school to a majority White school in the middle of high school; she detailed how her teachers in the new school admonished her for using African American Vernacular English (AAVE). She described how trying to master "Standard English" caused her frustration and made her feel as though she was less competent academically than her classmates; from her excerpt, it seemed that her teachers never considered praising her linguistic fluidity or finding ways to help her infuse both into classroom assignments. Instead, the teacher critiques maintaining the prevailing myth of Black inferiority in the schooling process, and made Alyssa feel culturally and intellectually deficient.

*10.5. Gatekeeping Grades and Opportunities (n = 7, 15%)*

Finally, seven women discussed how teachers controlled or limited their access to the grades they deserved or positive extracurricular opportunities. As with many of the discriminatory practices, the women were often unable to hold the teachers accountable for their behaviors. For instance, Dashawna, a 1st year Cameroonian student from a middle-class family, commented:

> I tried out for the cheer team my freshman year, and every girl looked the same. They were around the same height and had super light hair. I'm pretty tall, so I was towering above most of the girls. The try outs were over a couple of days, and I was by myself most of the time. I got a pretty good score, but I flipped to the back page and it was like, "We don't have a spot for you on the team." But when school started, there were girls on the team who didn't come to try outs. I emailed the coach and even though we had been emailing before, she didn't

reply anymore. I didn't get involved in any other sports in high school because I felt like I was going to have the same experience all over again.

Dashawna's early experience with cheer influenced her engagement in other extracurricular opportunities during high school. Even if there had been a logical reason that she did not make the team and the other girls did, the coach's lack of response indicated to Dashawna that it was more about her "not fitting in" than her cheer skills. Based on her explanation, she did not fit in because she was a tall, Black girl with short curly hair, compared to White girls on the team with "super light hair." In addition to teachers' gatekeeping extracurricular activities and opportunities, a few women discussed issues with unfair grading. Reflecting on her experience in an advanced placement (AP) class, Danielle, a 1st year Somalian American woman, said:

> She was the only teacher available, so, it was either drop the class or stay. Once, she took 30 points off my paper because I stapled it twice. Everybody else had feedback on their paper and I asked her, "Oh, Miss Cooper, why isn't there any writing on my paper?" I wanted feedback like everybody else. She looked through my paper for two seconds and said, "Your paper is stapled twice." I was like, "Is that really worth a 30 point deduction?" She caused me to get my first C ever. My parents called the school and the school was like, "Well, she's the teacher. We can't really do anything about it."

Danielle's example highlighted how, even with parental involvement, she was unable to get a grade adjustment. Although the experience did not factor significantly into her grade (she still earned an A), she recalled a number of similar incidents where the instructor treated her unfairly and made her feel unwelcome in the classroom. Overall, teacher gatekeeping did not derail the future of the young women in our sample because they were already on track to attend college and had significant financial, educational, or other instrumental support from parents. However, the consequences may be worse for Black girls who do not have strong and interconnected support systems to mitigate discriminatory teacher bias.

## 11. Anti-Racist Teacher Practices

### 11.1. Communicating High Expectations and Recognizing Potential (n = 13, 6%)

Rather than position the participants as "exceptions," thirteen women discussed how teachers affirmed their academic abilities in ways that humanized their experiences as high achieving Black girls. The instructors communicated their high expectations for the women and not only recognized their potential, but also pushed them to actualize success and reach goals that even the girls may not have believed they were capable of. In talking about her physics teacher, Indigo, a 2nd year African American student, shared:

> There were two physics awards and this guy, Chandler, was like a genius. We all knew it, so he won the physics award. I'm sitting there like, "Yeah, they're not going to call me." And then he called my name and I'm walking up there and you see all the little science nerd guys...the ones who went to college for engineering. And they're all just staring. But at the end of the day, I didn't think I deserved it. I thought that, once again, I was the one Black girl. Like I said, I was in his office every day. I used to stay after class with him...I used to email him...I was in there. So I know they know I'm a good person and they know that I work hard; but at the end of the day, I don't think I deserved that.

Indigo described her surprise at winning a physics award, even as she detailed the many ways that she demonstrated her commitment to learning. While her physics teacher recognized and rewarded her excellence, she was unable to see past her own internalized bias about whose hard work merited recognition. Thus, part of the significance of anti-racist teaching is that educators can help disrupt the deficit-based inner voices that some Black girls develop within mainstream schooling systems. Despite recounting the many

ways that she worked diligently during her time in high school, Indigo regularly discounted her academic potential and the significance of her hard work during our interview. Hannah, a 3rd year African American woman, shared:

> I was really good at school. I was always top of my class and I tried to get promoted with teachers and the principals and everything like that. So with like, being a Black woman, or being a Black girl—they were just like—you have to keep going...you have to keep going. You gotta make it out…so they set the precedent for how I got here. Because I always loved school and liked learning. I was really shy and quiet and always had my head in a book—I was always in the corner reading a book somewhere. Because from a very early age, I knew that to get out of my situation...school was the best option.

Hannah was the oldest of multiple children from a family that lived below the poverty line. During high school, she left to live with one of her aunts in a suburban neighborhood so that she could receive a better education and be the first in her family to attend college. She attested that her high school teachers recognized her potential and pushed her to "make it out." She conveyed the importance of the positive relationships that she built with several teachers and how they helped her realize her academic goals. As teachers embrace high achieving Black girls and express high expectations, they help normalize school settings as places to feel safe and seen.

*11.2. Challenging Racial Discrimination in the Moment (n = 4, 8%)*

Four young women detailed instances where teachers challenged racial discrimination in the moment. In order to help Black girls learn and to demonstrate a deep ethos of care and accountability, it is necessary that teachers and school administrators establish anti-racist school environments. This must include setting the precedent that racism is unacceptable and addressing racism when it happens. For instance, Akira, a 2nd year African American student, recalled:

> In the southwest part of the state…like in the mountains…it's very racist. I was on the basketball team and we played against this mostly White team. We had conflicts between us and them and we weren't treated very nicely and it was all because we're Black. We just kind of removed ourselves from the situation, but there wasn't really much we could do. Finally, we talked with our coach and they dealt with it. We didn't play against that school for a few years until things got sorted out.

Akira discussed her team's challenges with racism while playing a team that privileged whiteness; at first, the girls tried to manage the unfair treatment, but eventually decided to inform the coach since "there was nothing we could do about it." After she informed her coach, she recalled that the basketball team stopped playing that particular school for a few years. Thus, after learning about how the student athletes were being treated, school officials made the decision to remove them from the unsafe environment. In some cases, the teachers supported long-term efforts to disrupt discrimination. Brionna, a 1st year Sudanese student, talked about her year-long struggle during senior year to create a Black Student Union (BSU) at her school:

> After we got all the paperwork done and ran it by the club administrator, we had to go to the principal so he knows what clubs we have. We had this meeting with him and he was reading all the paperwork because the principal has to sign off to make it a club. He was talking to us about our goals with the club, and I remember he told me and my friend, "I don't think the students in this high school are ready for a BSU." We felt really invalidated. But we still kept going to make it work because our [teacher] sponsor was like, "Don't give up, we need this." It took from September to March to get him to sign it.

Brionna spent a significant amount of time during her senior year to start a BSU at her school, diverting time from her academic responsibilities to create an organization of support for Black students. She stated that Black students in her school deserved a space to talk about their experiences in the rural, predominantly White town and described the distress she felt when they encountered resistance from the principal. She also discussed the support that she received from the teacher who agreed to sponsor the BSU as a mentor, and how he diligently showed up to meetings with the principal until the form was signed and the club was officiated. Brionna's example highlighted how their student-led activism required educator support, as the principal possessed the authority to legitimize or subvert the students' efforts. The principal's language around "students not being ready for a BSU," highlights how racist ideologies shape school policies in school contexts—and the necessity of having teachers who are willing to supplant educational discourses that normalize colorblindness and privilege whiteness.

*11.3. Instilling Racial and Cultural Pride through Curriculum (n = 4, 8%)*

The final anti-racist category involved four women discussing teachers who tried to instill racial and cultural pride through their curriculum and classroom practices. The women's narratives demonstrated that teachers can challenge the prevailing myth of Black students' intellectual inferiority and shape Black students' perceptions of themselves as successful and agentic learners. Educators who privileged and prioritized Black students' racial and cultural histories and realities stood out as the women's "favorite teachers." This was also the one category where the women most often referenced Black teachers, highlighting the importance of same-race educators in increasing positive academic and social outcomes for Black students. Taylor, a 2nd year Nigerian woman, said:

> My calculus teacher and my psychology teacher in high school were the ones who really had an impact on me as far as letting me know who I was culturally. They were definitely like, "You're part of the Black community, and this is the history behind it. So this is what you have—you have to carry that torch." I never really understood that until I got to college, and I was like, "Wow." I remember when I came back and talked to them about it, they were like, "We couldn't really force you to understand. Your mind was going to open when it was going to open. But we just set the door there."

Taylor discussed the positive impact that two of her Black teachers had regarding her understanding of the legacy of educational excellence imbued in Black history. She also acknowledged that she understood the importance of her teachers' messages after matriculating to college, and notes that she went back and talked to them after her high school graduation. In many cases, the teachers' efforts continued to have a positive impact on the young women during college. Gabrielle, a 4th year African American student, shared:

> She did the kings and queens thing of highlighting our abilities and talents. I still talk to that teacher to this day and she's still doing the same thing—uplifting Black girls at that school and in the community and making them feel beautiful. I hold onto those people. It might not be every day, but I know that if I reach out to them, they're gonna be there.

Gabrielle demonstrated how one Black female teacher served as a role model for students, and tried to help prepare them to navigate a society that is filled with racial inequities and discrimination by communicating racial and cultural pride messages through poetry. The level of commitment shown by this and other Black teachers is commonplace; in addition, Gabrielle suggested that the teacher's affirming messages provided powerful imagery and aspirational ideals for them during high school. Without a contingent of Black educators within a school, Black students might lack the care, concern, and guidance these teachers offer.

## 12. Discussion

"Education is freedom work. In order for freedom to infuse and dominate the popular culture of our institutions, and the communities being educated in them, education cannot be understood as an exercise of privilege" (Morris 2019, p. 31).

The purpose of the current qualitative study was to explore how Black undergraduate women described the academic and social experiences they had with high school teachers, with a particular focus on misogynoir and anti-racism. To underscore the importance of dismantling racism and promoting racial justice in the classroom, the current reflective study provides a critical analysis of how Black undergraduate women experienced teacher support or marginalization during high school. Analogous to prior research on Black girls' experiences of racism and sexism in school settings (Carter Andrews et al. 2019; Evans-Winters and Esposito 2010; McArthur and Lane 2019), we found that participants recounted numerous forms of misogynoir in their experiences with teachers and school administrators. Many of the women described how educators policed their clothing choices and physical appearance, contributing to the type of adultification that can result in unfair disciplinary sanctions for dress code violations among Black girls (Anspach 2019; Blake and Epstein 2019; NWLC 2018). Women also received messages from teachers that their intellectual ability and hard work was an "exception" compared to other Black students, which created a harmful dynamic where the women felt pitted against same-race peers and felt more pressure to avoid mistakes and earn top grades (Anderson and Martin 2018). Finally, the findings illustrate that many of the participants' high school teachers overlooked the contributions of Black communities within U.S. history, effectively removing the opportunity for the girls to "see themselves" in the curriculum.

Although discriminatory practices were far more common in the women's narratives, a few participants discussed anti-racist experiences with teachers during high school that challenged the types of implicit and explicit schooling practices that harmed the women in our study. These women encountered supportive and affirming teachers (e.g., warm demanders; Ware 2006) that responded to the girls' potential with high academic expectations and an ethos of care. In addition, some of the women discussed how teachers called out racial bias in the moment or provided long-term support to challenge discrimination. This demonstrated to the young women that these teachers refused to be complicit in the dominant ideology of colorblindness (i.e., refusal to acknowledge the costs and benefits associated with one's racial and cultural identity; Ullucci and Battey 2011), and instead, took individual action to counteract racism and support their students. Black teachers, in particular, challenged the hegemonic effects of whiteness and anti-Blackness (Gist et al. 2017; McArthur and Lane 2019) by infusing cross-cultural perspectives and Black history within the curriculum. In addition, the women recounted how Black educators, in particular (while few in number), offered not only academic support, but also guidance on how to navigate racism and inequality. In sum, some educators demonstrated a commitment to their Black female students and promoted equity in the classroom in ways that made the young women feel seen, respected, and valued.

### 12.1. The Impact of Teacher Misogynoir on Black Girls' Schooling Experiences

Consistent with prior research (e.g., McArthur 2018; Morris 2016a), our findings revealed that the young women encountered a number of issues (e.g., policing appearance and tone of voice, tokenization, and unfair grading policies) that challenged their engagement and sense of belonging in high school. For instance, while all the women in our sample were enrolled in highly selective undergraduate institutions, fourteen women discussed how teachers communicated low expectations of their academic ability. The young women who attended predominantly White or racially diverse high schools, often had teachers who negatively compared Black students' performance to White or Asian students. The young women who attended predominantly Black high schools tended to have

teachers who had lower expectations of Black students in general. Andrews and colleagues (2019) suggested that limited awareness from teachers, regarding Black girls' unique challenges in school, perpetuates the mischaracterization of their attitudes, abilities, and achievements. Their participants described many of the same toxic racialized and gendered experiences with adults and peers that we found in the current study (i.e., lower academic expectations, marginalization of Black female athletes, and being hypersexualized and adultified). The high school-aged girls discussed the double standards in constructions of femininity, such as experiencing stricter enforcement of the dress code than White female students. Within the current study, the young women's reflections revealed how day-to-day interactions with teachers influenced how they viewed themselves, experienced school, and understood their future academic potential.

While some of the young women's teachers fed into derogatory narratives about Black student achievement, our participants also described a form of exceptionalism and tokenization relegated to high-achieving Black students (Archer-Banks and Behar-Horenstein 2012; Carter Andrews 2009). For example, Alexa, a 3rd year African American student, described how a number of her instructors said that a close Black male friend of hers would "end up in jail." After he was incarcerated during high school, she used her friend's negative experiences with teachers as a guiding force to continue her path towards becoming a successful early childhood educator. She believed that these teachers played a key role in derailing his chances for future educational and occupational success, but she also described how her high performance placed her outside of these critiques. While prior researchers highlight how Black girls are often positioned as undisciplined in their academic habits and unequivocally misaligned with school norms (Carter Andrews et al. 2019; Froyum 2010), our findings suggest that teachers also harm Black girls in school by encouraging a maladaptive form of perfectionism to "prove" their worth as students (Anderson and Martin 2018). As the number of identified gifted and talented Black girls increases, there is a strong and urgent need for teachers to be prepared to address their social-emotional needs (Evans-Winters 2014). Researchers have noted that maladaptive behaviors associated with perfectionism can contribute to depression and anxiety (Anderson 2020), and while beyond the scope of the current study, 11 (22%) women indicated that they struggled with mental health issues due to academic stress. While many of the women in our study acknowledged that their teachers spoke of them as highly ambitious learners, they still battled the non-affirming cultural norms of their schools, including a felt sense of pressure to be all-knowing and perfect (Carter Andrews et al. 2019).

### 12.2. Situating Black Girls as Knowers and Change Agents

Our findings also revealed that it is important for educators to transform educational contexts into spaces that help Black girls recognize and critique oppressive narratives about their identities (Jacobs 2016; Smith-Evans et al. 2014). When implemented correctly, anti-racist practices that incorporate gender and sexism have the potential to free Black girls from policing their own potential by learning about how identity-based oppression influence interpersonal relationships (i.e., teacher–student relationships) and larger social structures (i.e., schooling contexts). Consistent with CRF's focus on Black women's multiple ways of knowing, the women's reflections called attention to how Black girls are required to develop their personal beliefs and values about their identities, while also being aware and conscious of how bias and discrimination influence their schooling experiences. For example, teachers' tokenization placed an unfair burden that presumed the young women could attest to the experiences of all Black people, which led to feelings of social isolation, alienation, and invisibility. Morris (2016a) demonstrated the harm that can befall Black girls (i.e., in-school suspension, expulsion, and sexual objectification) when they are unable to navigate and survive in schools that require this multiple consciousness. Still, while our participants described how they had to shift between their personalized identities and the expectations of primarily White teachers and school administrators, their presumed identities as "good students" protected them from many of the

harsher disciplinary actions in schools against other Black girls (Hines-Datiri and Carter Andrews 2020; Wun 2014). Similar to prior work (e.g., Evans-Winters 2005, 2014), our participants were likely able to negotiate misogynoir and discrimination from teachers and achieve academic markers of success because they received adequate support from their families and communities.

Yet, very few of the women recalled moments where their teachers specifically engaged in critical and caring pedagogical acts to empower them to understand themselves better as Black girl knowers (e.g., Greene 2020). A small number of the women described teachers who provided a diverse and more accurate rendering of history (including reciting poems that encouraged racial and cultural pride); but before college, none of the young women encountered classroom spaces that centered on their realities and helped them define their experiences (McArthur and Lane 2019; Nyachae 2016; Price-Dennis et al. 2017). Scholarship on identity development highlights that Black women and girls are cognizant of how their female identity often differs from traditional or mainstream notions of femininity (Archer-Banks and Behar-Horenstein 2012; Carter Andrews et al. 2019; Mims and Williams 2020), and our findings underscore that critical pedagogical practices offer one avenue for teachers to significantly improve Black girls' schooling experiences. Specifically, they can draw on anti-racist curriculum to highlight how Black women and girls' experiences are different from the experiences of men of color and those of White women (Evans-Winters and Esposito 2010). Educators can offer assignments and group activities that focus on systemic oppression at the intersections of race, class, and gender (i.e., White male patriarchy and racist oppression) to counter the disillusionment that Black girls may experience in schools (McArthur and Lane 2019; Morris 2019). Our participants seemed to have failed to realize that this should or could have been a possibility for their high school education, an unfortunate outcome of colorblind curriculum and school policies that downplay the significance of race and gender, while propagating a stance of neutrality, objectivity, and White middle-class normativity (Evans-Winter 2014). Teachers who offer complex understandings of social structures situate Black girls as curators of knowledge and prepare them to resist misogynoiristic perspectives.

### 12.3. Demonstrating a Politicized Ethic of Care among High School Educators

Prior scholars highlight the narrow and attenuated nature of many mainstream schooling curricula (e.g., Charles 2019; Dee and Penner 2017), which was true for most of the women in our sample, as well. Moreover, some women recounted how high school teachers placed the burden on their shoulders to explore and explain racism to their classmates, a tokenizing experience that can be traumatic for students who are also learning. Evidence suggests that anti-racist curriculum draws on students' culture heritage as a basis for classroom practice and curricular content, which can contribute to critical self-understandings that bridge family, school, and community experiences (Dee and Penner 2017; Sealey-Ruiz and Greene 2011). Yet, most of the young women's teachers offered race neutral and colorblind versions of history that erased the harm wrought from racism, capitalism, patriarchy, and colonialism. Similar to prior research (e.g., Olusoga 2015), their teachers seemed to try to encapsulate the richness and abundance of Black history in a single month (i.e., February), or they presented isolated historical understandings that disconnected particular events from the larger socio-political landscape. For instance, students who lack a comprehensive understanding of the enduring effects of enslavement and Black disenfranchisement in the U.S., might believe that George Floyd's murder alone, ignited the 2020 Black Lives Matter protests. Yet, as Brown (2020) pointed out, Black America is still recovering from the gruesome lynching of Emmett Till. Mainstream whitewashed history downplays the harm enacted against Indigenous, Native peoples and Africans in the Americas; it limits students' understanding of how historically marginalized communities have negotiated oppressive spaces, enacted resistance, and fought against marginalization (Charles 2019; Sealey-Ruiz and Greene 2011). Thus, our findings align

with other scholarship suggesting that teachers must facilitate classroom spaces that challenge students to examine personal assumptions around race and deconstruct harmful stereotypes of anti-Blackness and misogynoir (Ware 2006; Wells et al. 2016).

Finally, we believe that schools should elevate the expertise and knowledge of Black educators and scholars who have been engaging in anti-racist teaching for decades (e.g., Hanna 2019). Some scholars document how Black teachers can create classroom communities that focus on Black boys at the expense of Black girls (e.g., Brown 2011) or employ forms of systemic emotional and behavioral control that harm Black girls' development (i.e., stifling the girls' attitudes, making girls emotionally accountable for their younger peers, and demanding sympathy for adult authority figures; Froyum 2010). Yet, our findings align with a larger body of work demonstrating the benefits of having Black educators, specifically, in the classroom (e.g., Greene 2020; Hanna 2019; Lane 2017). The women in our study described how Black educators created cultural bonds with them through interpersonal interactions that highlighted their academic strengths and encouraged them to embrace their #blackgirlmagic (Lane 2017; McArthur 2019). Our participants also recounted how they benefited from the access to same-race role models and potential mentors; Black teachers served as resilient protectors against misogynoir by uplifting them during high school and reminding them of their potential. Our findings demonstrate how Black teachers actively disrupted and reduced the young women's exposure to harm during high school through their ability to respect and engage with them. Future research should consider how educators nurture a sense of sisterhood and actively promote cultural pride among Black girls within anti-racist curricula (Aston et al. 2017; Gist et al. 2017).

## 13. Limitations

Despite the strengths of the current study, there are a few limitations worth noting. For example, we used a reflective qualitative approach with Black undergraduate women regarding their high school experiences, rather than talking with Black girls currently in high school. This was an intentional theoretical and methodological choice to capture the women's significant memories about teachers after they had successfully graduated from the school; however, we might have gained different insights if we had collected data with Black girls currently enrolled in high school. Furthermore, while the high school years and late adolescence are a critical period for identity development (Carter Andrews et al. 2019), many Black children start thinking about and making sense of their social identities as early as elementary and middle school (Dulin-Keita et al. 2011). Participants in the current sample recounted several experiences with racism and sexism in earlier grades, but we decided to focus the present paper on the women's high school years. Developmental scholars will gain important information by examining the beliefs and attitudes Black girls hold about their social identities, as well as the processes that contribute to their learning about racism and sexism during early adolescence and into emerging adulthood (Mims and Williams 2020).

In addition, a growing number of studies document the unique racialized experiences and perspectives among Black African immigrant students (e.g., Daoud et al. 2018; Hernandez 2012; Mwangi and Fries-Britt 2015). While all the women in the current sample attended schools in the U.S., and thus, seemed to share certain types of racialized and gendered experiences with high school educators, it is important to call attention to within-group diversity in ethnically diverse Black students' schooling experiences. If we had coded for qualitative differences in the misogynoiristic experiences of African American women and Black African immigrant women, the data might have revealed nuanced understandings of how Blackness is defined in America and Black African immigrants' social positioning in U.S. classrooms (Mwangi and Fries-Britt 2015). Researchers affirm that girls from Black African households receive unique socialization from parents and family members regarding education and sociological understandings of race (Kiramba et al. 2020), which may have informed their interactions with teachers and perceptions of

their school settings. Still, much of this work looks at foreign-born Black immigrant students (for review, see Daoud et al. 2018) and the current sample included women who were born and raised in the U.S. Moreover, while the young women in the study may have possessed divergent beliefs about how race and gender operate, their teachers may have been less likely to know about the economic, social, and cultural distinctions among ethnically diverse Black girls in their classroom (Tatum 2017). Thus, their interactions were likely informed by homogenizing understandings about Blackness and race.

In addition, the study would have been strengthened if we had data from teachers to explore their perceptions of the young women during high school and triangulate our findings. Evidence suggests there may be a disjuncture between teachers' perceptions of Black girls and the students' personal goals and actions (Kelly 2018; Neal-Jackson 2018); however, this remains an understudied area of research in relation to Black girls in high school and the teachers who educate them. Finally, we did not ask explicitly about anti-racist teacher practices during the interviews, which might explain the discrepancy in how many more discriminatory events that the participants recalled compared to race-conscious educational practices. However, the protocol question asked about teacher experiences broadly; thus, this discrepancy in the women's narratives draws attention to the prevalence of racism and sexism in their high schools, and underscores the need for systemic transformation in how we address white supremacy in educational settings (Arellano and Vue 2019). Future research might also consider exploring how teacher and school administrators' responses shift with older youth (i.e., how do anti-racist strategies change with time as students develop stronger critical thinking skills?).

## 14. Implications for Scholarly and Educational Praxis

The present study demonstrates the importance of reframing teacher practices to address Black girls' experiences of misogynoir in high school and has several implications for scholarly and educational praxis. First, the women's narratives denote the importance of teachers seeking feedback and listening to the knowledge and meaning-making of Black girls when they speak about their schooling experiences. Many participants doubted their intellectual ability in the classroom, even as they earned high grades or demonstrated academic excellence. Educators must intentionally recognize and foster Black girls' academic potential, while also being mindful that they are not nurturing perfectionist tendencies (especially for Black girls who are the only or one of few in the classroom; Anderson 2020). Teacher preparation programs should be at the forefront of attending to critical discourse that challenges systemic oppression based on negative images and perceptions of Black girls. To combat misogynoir and nurture Black girls' oppositional gaze, teachers should do the following: (1) acknowledge and be willing to address their own biases; (2) develop tools that can assist with identifying and intervening when Black girls are being harassed or discriminated against; (3) constantly evaluate data and outcomes of Black girls at the classroom and school-wide level to work towards collaborative change; and (4) be an advocate for Black girls who are learning to utilize their oppositional gaze as a form of resistance. Instead of critiquing Black girls, these practices focus on teachers' reflectiveness and accountability in how they interact with their students.

Second, the findings reveal the positive impact of increasing the number of Black educators in the classroom. Regardless of their high school racial composition, participants who built meaningful relationships with Black teachers found supportive role models and spoke about their sense of cultural connectivity in the classroom (Gist et al. 2017). The women discussed how Black teachers had a better understanding about the socioemotional experiences of Black students in the school, which allowed them to push the women's intellectual identities as students. These educators were willing to resist and combat the academic marginalization of Black students by infusing their curriculum with Black history and anti-racist pedagogy. While beyond the scope of our study, teachers can also support Black girls' critical ways of knowing outside of the classroom through affinity groups and programs that encourage positive interaction among Black girls (Watson

2016). By providing a space in which Black girls are not being evaluated on their responses, there will be opportunities for growth and identity development that are not necessarily available in the classroom.

Third, the women's discussions elucidated the importance of decolonizing mainstream curriculum to include lesson plans that address examples of negative stereotypes in media while allowing students to openly discuss these types of portrayals in class (Kelly 2018; Price-Dennis 2016; Price-Dennis et al. 2017). For example, adding references to music genres and social media can foster more academic engagement and enjoyment in the classroom by utilizing contexts and information that are relevant to Black girls' lives (Price-Dennis 2016). Educators must consider how to actively diversify and decolonize the canons and histories in their classroom communities. Beyond curriculum, teachers and schools could also acknowledge annual holidays outside of the Christian tradition (e.g., Eid al-Fitr, Kwanzaa, Juneteenth, and Rosh Hashanah), and emphasize open conversations about systems of oppression and power (Lawrence 2005). Our findings are relevant to all students because they are members of a multicultural society and world, and educators must help students examine how their social identities (e.g., race, gender, social class, and sexuality) impact the learning space and dynamics in the classroom.

Finally, our findings are particularly important in thinking about incorporating intersectional anti-racist practices into school-wide policies. This should involve intentional recruitment and retention of educators of color, as the K–12 educational workforce (teachers and principals included) is still majority White (U.S. Department of Education 2016). School administrators should provide professional development opportunities, consistent evaluation, and mentoring for teachers who are committed to anti-racism and equity. There are a number of scholarly and community experts who specialize in anti-racist leadership who can lend support to transform schooling contexts (Superville 2020). In addition, school leaders must be open to addressing racist incidents that occur inside and outside of the school with students and model how to engage in proactive dialogue around race, colorblindness, and anti-racism. Educators and school leaders, particularly those who are White, must not assume that they know the best ways to support Black students; instead, they should be open to seeking and incorporating feedback from parents and students about ways to improve the school's racial climate and classroom dynamics (Schniedewind and Tanis 2017). Finally, anti-racist schooling systems do not occur in a vacuum—teachers and principals need support from superintendents and school boards to enact district-level change (Irby et al. 2019). Anti-racist work is a long-term, and critical commitment if educators want to create learning contexts that offer students equitable opportunities to thrive. We need anti-racist school systems because we cannot expect exceptional teachers to enact systemic-level change, just as we cannot hope that Black girls chance upon a dedicated teacher who recognizes their potential and helps them achieve the next step in their educational journey.

**Author Contributions:** Conceptualization, S.L., P.B., and T.P.; methodology, S.L.; software, S.L.; validation, S.L. and N.W.; formal analysis, S.L., P.B., T.P., and N.W.; investigation, S.L., M.W., and D.J. resources, S.L.; writing—original draft preparation, S.L., N.W., P.B., M.S., and W.M.; writing—review and editing, S.L., N.W., P.B., M.S., W.M., and T.P.; supervision, S.L.; project administration, S.L. All authors have read and agreed to the published version of the manuscript.

**Funding:** This research received no external funding.

**Institutional Review Board Statement:** The study was conducted according to the guidelines of the Declaration of Helsinki, and approved by the Institutional Review Board (or Ethics Committee) of the University of Virginia (protocol code 2897 approved on 10 September 2019).

**Informed Consent Statement:** Informed consent was obtained from all subjects involved in the study.

**Data Availability Statement:** Abridged versions of the data presented in this study are available on request from the corresponding author.

**Acknowledgments:** We would like to deeply thank the Black undergraduate women who shared their beliefs, perspectives, and experiences with us—this manuscript would not be possible without your insight and willingness to share.

**Conflicts of Interest:** The authors declare no conflict of interest.

**Appendix A**

We have two main goals in talking with you: (1) to discuss the messages you received while growing up about your identity as a Black girl, and (2) to talk with you about how your media consumption relates to your ideas about your Black woman identity. These questions are meant to dive into the socialization messages that you received, both implicitly and explicitly, during your childhood and adolescence. We are interested in the types of messages you may have received around topics like those included on this sheet [Hand them sheet with general examples].

1. What words or characteristics would you use to describe yourself as a Black woman?
2. I'd like you to tell me a story about an important positive experience in your life that related to your identity as a Black girl.
   a. How did this experience affect you? Make you feel?
   b. Why does this experience stand out to you?
3. I'd like you to tell me a story about an important experience in your life that presented a challenge related to your identity as a Black girl.
   a. How did this experience affect you? Make you feel?
   b. How did you make decisions on how to deal with or resolve this challenge?
4. During K–12, did you receive messages about being a Black girl from people at school? This includes peers, teachers, or school figures.
   a. What did you think about these messages?
   b. Do you recall receiving any messages that you disagreed with?

   **General Examples:**

- Hair
- Body shape or body image
- Colorism—comments about skin tone
- Correct ways to conduct yourself as a girl/woman
- What to do if you encounter race or gender discrimination
- How to interact in interracial friendships or relationships
- Gendered expectations—role in the family as a girl/woman
- How to respond to sexual harassment/assault (including cat-calling)
- Future educational or occupational goals

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
