# Peer review of "A Qualitative Study of Black College Women’s Experiences of Misogynoir and Anti-Racism with High School Educators"

_socsci, doi:10.3390/socsci10010029_

Round 1

Reviewer 1 Report

Thank you for the opportunity to review this interesting and important piece on how black college women reflect back on their high school experiences.

I enjoyed reading the manuscript but there are a couple of things that I would like to suggest.

This paper is rich in empirical material and has an inductive approach. In the concluding discussions, the depth of theoretical discussion is lacking. With that said, the concluding discussions has great practical implications. One way to deepen the theoretical discussion can be to connect the empirical material and the concluding remarks with the concept of "oppositional gaze" more explicitly. My understanding of your argument is precisely the need of teachers who encourage oppositional gaze, instead of practicing misogynoir. 

The description of your participants who identified as African should be elaborated more. 22 identified as African and you state that they are "from" different countries. I assume that they all migrated to the US before starting high school (and that their entire high school education is from the US)? The age of migration/years in the US normally affect the way you see your experiences of racisms and therefore it should be specified how long the 22 interviewees have been in the US. There are previous studies which shows the different experiences of racisms and exclusion that African American population and the immigrant African population face. This is why I believe that information on when they migrated and how long they have been in the country is relevant.

Related to this, did you see any differences in the reflection of experiences among those who identified as African American and African? Have you found any patterns/differences in the ways of reflection between the two groups while you coded the interviews? I would like to see some explanation on this, why you treat all in one group as "black college women".

Author Response

The description of your participants who identified as African should be elaborated more. 22 identified as African and you state that they are "from" different countries. I assume that they all migrated to the US before starting high school (and that their entire high school education is from the US)? The age of migration/years in the US normally affect the way you see your experiences of racism and therefore it should be specified how long the 22 interviewees have been in the US. There are previous studies which show the different experiences of racism and exclusion that African American population and the immigrant African population face. This is why I believe that information on when they migrated and how long they have been in the country is relevant.

  • We appreciate this feedback, and we added a sentence about what we know of how long each woman spent in her parent’s country of origin (p. 7). To the reviewer’s other question, all of the women were U.S. citizens. We provided additional details in Table 1 on each woman’s ethnic identification, and we agree with the author about the potential divergence in African American and immigrant Africans’ understanding of race and experiences with racism.

There is no explanation of the racial demographics at either school and there should be. 

  • We added this to the Participants section in the Methods (i.e., “Twenty-one women attended a public institution in the Midwest (5% Black / African American) and the remaining 29 attended a selective, public institution in the Southeast (6% Black / African American) (p. 6).

This paper is rich in empirical material and has an inductive approach. In the concluding discussions of implications, the depth is lacking. One way to deepen the theoretical discussion can be to connect the empirical material and the concluding remarks with the concept of "oppositional gaze" more explicitly.

  • We think the author has valid feedback regarding the discussion. Based on this, we added the following: 
    • “Teacher preparation programs should be at the forefront of attending to critical discourse that challenges systemic oppression based on negative images and perceptions of Black girls. To combat misogynoir through oppositional gaze, teachers should do the following: 1) acknowledge and be willing to address their own biases, 2) develop tools that can assist with identifying and intervening when Black girls are being harassed or discriminated against, 3) work to understand concepts such as code-switching and behavioral norms from various backgrounds as not to punish what may be misinterpreted as misbehaving, 4) constantly evaluate data and outcomes of Black girls at the classroom and school-wide level to work towards collaborative change, and 5) be an advocate for Black girls who are learning to utilize oppositional gaze as a form of resistance. Instead of critiquing Black girls, these practices seek to understand their actions and intersectional identity through personal accountability and reflection.”  (p. 22-23)

The section on limitations is useful and could be expanded slightly. The inclusion of African young women without explanation is a limitation, especially since most of the literature review describes African American adolescent girls.

  • We added this to the Limitation section (p. 22).
    • “Also, a growing number of studies document the unique racialized experiences and perspectives among Black African immigrant students (e.g., Daoud et al., 2018; Hernandez, 2012; Mwangi & Fries-Britt, 2015). While all of the women in the current sample attended schools in the U.S., and thus, seemed to share certain types of racialized and gendered experiences with high school educators, it is important to call attention to within-group diversity in ethnically diverse Black students’ schooling experiences. If we had coded for qualitative differences in the misogynoiristic experiences of African American women and Black African immigrant women, the data might have revealed nuanced understandings of how Blackness is defined in America and Black African immigrants’ social positioning in U.S. classrooms (Mwangi & Fries-Britt, 2015). Researchers affirm that girls from Black African households receive unique socialization from parents and family members regarding education and sociological understandings of race (Kiramba et al., 2020), which may have informed their interactions with teachers and perceptions of their school settings. Still, much of this work looks at foreign-born Black immigrant students (for review, see Daoud et al., 2018) and the current sample included women who were born and raised in the U.S. Moreover, while the young women in the study may have possessed divergent beliefs about how race and gender operate, their teachers may have been less likely to know about the economic, social, and cultural distinctions among ethnically diverse Black girls in their classroom (Tatum, 2017). Thus, their interactions with the young women were likely informed by homogenizing understandings about Blackness and race.”

Reviewer 2 Report

I applaud the authors for this essential work! This paper was wonderfully written and I am in strong support of this manuscript. In "A Qualitative Study of Black College Women’s Experiences of Misogynoir and Anti-racism with High School Educators" the authors share critical insights about how Black college women understand their earlier school experiences. The authors firmly ground their study within existing literature in education and Black womanhood across disciplines. The methods section was clearly written and may serve as a roadmap for other qualitative scholars. The findings section included illustrative quotes and sound interpretations, which supported their analyses. Overall, in describing discriminatory and anti-racist teaching practices, the authors greatly expanded upon what we already know about the schooling experiences of Black girls. 

Author Response

Thank you for your feedback - we incorporated a number of changes to strengthen the conceptual framing and findings section of the manuscript. This review was helpful in reminding us of the importance of this work!

Reviewer 3 Report

Overall, the aims of the study will make a significant contribution to the field particularly in relation to discrimination, misogynoir, and anti-racist work in educational settings by way of Black girls’ experiences in schooling.

Literature Review

The import of the two concepts, misogynoir and anti-racist, is significant in the field of education generally and the particularly in the experiences of Black girls more specifically. However, these terms genealogically, are not fleshed out for readers. I would recommend an in-depth review of the term misogynoir in relation to how this provides a unique perspective on schooling experiences of Black girls. Currently, as it reads, the section on misogynoir more so emphases both exclusionary discipline and discrimination in punishment among Black girls, rather than how misogynoir informs these practices. The work of Moya Bailey would be helpful here. To dive into how misogynoir is the underbelly to this analysis would provide nuance for present studies on Black girls and their schooling experiences to show the harmful consequences of such punishment that frames their experiences. The concept of anti-racist needs to be fleshed out and show why an anti-racist frame is unique from diversity and inclusion frameworks/initiatives. This is a powerful term currently and it is important to illustrate explicitly what it offers as a concept historically and why it is useful here. I would review work that explains its origin by Angela Davis.  

I would encourage the authors to think through the significance of dehumanization. What is your definition of dehumanization? What are its origins? Are the negative social interactions with teachers a byproduct of the dehumanization? Connie Wun’s work would be useful to think through dehumanization with an anti-blackness lens in the educational experiences of Black girls in schooling. Also, you provide this significance of humanizing work in your discussion section under “Humanizing Black Girls in the Classroom through Anti-Racist Curriculum.” Bring this literature up to the politicized ethic of care section to strengthen it.

Method/Methodology

The description of how this study was conducted and the process to secure inter-rater reliability among the authors is thoroughly detailed. I would have liked to see a table that described participants in the study including the neighborhood context of their schools (city, rural, or suburban), SES, IB/AP/honor roll, the college each participant was enrolled in and whether or not they were private and public, etc. This type of information in a table would help the reader contextualize the findings. I was curious about why authors chose to interview students enrolled in college about their high school experiences, instead of talking to high school students. This is discussed in the limitations section, I would bring up this explanation to the methodology section where there can is a description of why it is important to talk to Black girls enrolled in college about their discriminatory experiences in high school to show how they are not protected from this type of discipline even though they went on to achieve success by enrolling in college.

Results

I did not find the tables with the summary of discriminatory teacher practices and anti-racist teacher practices particularly useful because you provide in-depth analyses of these incidents in your findings. Your findings do not necessarily analytically engage misogynoir as an analytic. Instead, you provide descriptions of the different forms of discrimination participants went through. In regard to the anti-racist work, most of the data pulled shows Black girls engaging in this work rather than their teachers. The teachers were verbally supportive but lacked in transformative action on behalf of their students.

Conclusion

The discussion section provides the reader with the significance of this study and larger conversations necessary to transform the schooling experiences of Black girls. A lot of this information would strengthen the findings section in terms of the analysis and supporting claims made. Authors provide interactional level interventions to remedy discriminatory and misogynoir practices, but what are the institutional level changes that can be made to change practices in school?   

Author Response

Introduction

The import of the two concepts, misogynoir and anti-racist, is significant in the field of education generally and the particularly in the experiences of Black girls more specifically. However, these terms genealogically, are not fleshed out for readers. I would recommend an in-depth review of the term misogynoir in relation to how this provides a unique perspective on schooling experiences of Black girls. Currently, as it reads, the section on misogynoir more so emphases both exclusionary discipline and discrimination in punishment among Black girls, rather than how misogynoir informs these practices. 

  • We appreciate this feedback and made sure that we clearly outlined definitions of misogynoir and anti-racism from the first time that they were mentioned in the text (p. 2). While it was beyond the scope of the current study to provide a scoping review of misogynoir and anti-racism, we did include several new citations about Black women and girls’ intersectional experiences; Ford et al. 2019 - Engaging and empowering gifted Black girls using multicultural literature; Lindsay-Dennis, 2015 - Toward a culturally relevant research model focused on African American girls; Wing - Critical race feminism).
    • “This reflective approach allowed us to engage the women in questions about how teachers helped or hindered them throughout high school, focusing especially on Black women’s discussions of misogynoir (i.e., the ways in which racism and sexism intersect to produce racialized gendered violence and harm against Black women and girls; Trudy, 2014) and anti-racist teacher practices (i.e., efforts that go beyond education reform, to transform the structural inequities that maintain racism; Lynch et al., 2017) among secondary school educators. We draw on critical race feminism (CRF) as a theoretical tool to analyze Black women’s racialized and gendered experiences with their high school teachers (Evans-Winters & Esposito, 2010).” (p. 2)
  • We also tried to provide more analysis of how misogynoir is an ideological concept that informs specific practices (as the reviewer pointed out) that harm Black girls in educational settings. 
    • Over the past few decades, research concerning educational interventions that challenge racial oppression has grown, along with awareness among lay and scholarly audiences of the urgent need for such interventions (e.g., De Lissovoy & Brown, 2013; Gillborn, 2005; Ladson-Billings, 2016). Yet, a growing body of evidence demonstrates that many of Black girls’ school-related challenges involve misogynoiristic practices from teachers and school officials (Carter Andrews et al., 2019; Davis, 2020; Morris, 2016; Neal-Jackson, 2018; Watson, 2016). Misogynoir refers to the ways in which racism and sexism intersect and perpetuate harm against Black women and girls through interpersonal encounters and institutional structures (Trudy, 2014). Understanding the functionality of misogynoir as an ideological construct can help scholars and educators contextualize and address the innumerous ways that Black women and girls are subject to inequitable and harmful treatment in educational settings and broader society (Bailey & Trudy, 2018).” (p. 3)
  • We were using work from Trudy (a colleague of Moya Bailey, but added a specific citation with Bailey - they have an article on citational erasure that is useful, but doesn’t speak as much to misogynoir in educational settings). We believe a strength of the current study is drawing on this literature and incorporating misogynoir as a term / concept in thinking through Black girls’ schooling experiences. 

There isn’t an overarching theory to focus the study design.

  • We appreciate this feedback and added a section on Critical Race Feminism as a guiding theoretical framework for the current study (p. 2). While we did not have the space to provide a genealogical review of CRF, we explained how it derived from critical race theory, introduced the main tenets, and offered a few explanations of how it undergirds the current investigation. Our interview questions and conceptual framing were already informed by Evans-Winters’ Black Feminism in Qualitative Inquiry as well as intersectionality theory, so this seemed like a strong conceptual fit as an overarching theory. We also added citations with studies on Black girls that incorporated CRF (e.g., Lindsay-Dennis, L. (2015). Black feminist-womanist research paradigm: Toward a culturally relevant research model focused on African American girls. Journal of Black Studies, 46(5), 506–520).

I would encourage the authors to think through the significance of dehumanization. What is your definition of dehumanization? What are its origins? Are the negative social interactions with teachers a byproduct of the dehumanization? Connie Wun’s work would be useful to think through dehumanization with an anti-blackness lens in the educational experiences of Black girls in schooling. -Seanna

  • We appreciate the reviewer’s concerns about our use of the word dehumanization. While we believe that it is useful in describing many of the women’s experiences with teachers in the current study, we also opted to remove it from the manuscript at this time (it was used in 6 places, which we revised). Given that we already spent some time expanding on misogynoir and anti-racism, as well as critical race feminism as a theoretical framework - we did not want to introduce additional theory on the roots of dehumanization. However, we did include - Wun, C. (2014). Unaccounted foundations: Black girls, anti-Black racism, and punishment in schools. Critical Sociology, 1-14. https://doi.org/10.1177/0896920514560444 - as a new citation because (as the reviewer suggested) the author does an excellent job explaining how anti-black racism affects school policies and practices that harm Black girls.

Pages 2-4 primarily focus on adolescent girls and should actually focus on both adolescent and young adult/college age women. This will provide a more balanced perspective about the study participants.

  • We included a few more sentences about college aged women, but since the focus was on the young women’s experiences during high school, we chose to keep the primary focus on empirical literature with Black girls in K-12 (and mostly in high school). We did include a few citations on Black college women when relevant (e.g., Patton & Croom, 2017). Based on feedback from another reviewer, we revised the Introduction to make a clear case about why we used a reflective approach with a sample of undergraduate women (p. 2).

Also, some of this content could be cut down especially the focus on girls being arrested in high school since that was not the focus or noted in the findings for the current study.

  • We only mentioned the juvenile court system in one sentence in the Introduction (i.e., and represent one of the fastest growing youth populations in the juvenile court system (Blake et al., 2015; Gibson et al., 2019), but we removed it for the revision. 

The authors include three sections in the background about anti-racist praxis, politicized ethic of care and restorative educational justice that are not clearly defined/explained as perspectives and/or content to inform the study and design, etc. One suggestion would be to add an overview – a few sentences before the paragraph about anti-racist praxis to explain the purpose of this information and why it’s needed.

  • The subheading “Moving towards Anti-Racist Educational Praxis” (p. 4) was intended to serve as a transition between our discussion about misogynoir and anti-racism. However, we made this section more specific and outlined the main points of the prior two subsections (i.e., Developing Black Girls’ Critical Ways of Knowing, Building a Politicized Ethic of Care among High School Teachers, and Dismantling School Structures that Harm Black Girls). We changed the order of to move from Black girls → Teachers → School Structures (micro to macro). 
  • We also added the third subsection about Transforming School Structures to highlight three levels of anti-racist intervention. We provided definitions for each subsection that directly tied back to anti-racist educational practices, which may have been a missing link in the initial submission. We hope that this new framing also attends to reviewer comments about improving how we discuss the structural implications of this work for schools.
    • “To challenge hegemonic educational systems, it is necessary to adopt anti-racist strategies that maximize the educational opportunities and socioemotional well-being of Black girls and women (Neal-Jackson, 2018). Although it is the responsibility of educators and school leaders to address how racism influences the day-to-day routines in schools, researchers suggest that Black girls and their families learn to do their best to assimilate and navigate the cultural demands of schools (Holland, 2012; Lewis & Diamond, 2015). Below, we review three key domains of anti-racist educational praxis that – when integrated – should catalyze more just and equitable schooling environments for Black girls. First, we review literature on cultivating an “oppositional gaze” (hooks, 1992; Jacobs) among Black girls to help them interpret the misogynoiristic experiences they have in school within a broader understanding of the historical and institutional contexts of racism and sexism. Second, we highlight the role that teachers can play by investing in Black girls’ lived realities and implementing anti-racist pedagogical practices (Greene, 2020; Lane, 2017) .Third, we describe how equity-oriented school leadership are critical for disrupting school structures that marginalize and impede the academic success of Black girls (and students of color more broadly).” (p. 4-7)

This type of information in a table would help the reader contextualize the findings. I was curious about why authors chose to interview students enrolled in college about their high school experiences, instead of talking to high school students. This is discussed in the limitations section, I would bring this up sooner.

  • We appreciate this feedback, and added the following to the Introduction:
    • “Furthermore, for some young Black women more than others, high school educators play a critical role in regards to advanced degree attainment, career choice, and lifetime earnings (Patton et al., 2017). Yet, less research attends to how Black girls navigate high school and achieve their educational goals in spite of school-based marginalization (for notable exceptions, see Anderson, 2020; Carter Andrews et al., 2019). To build upon extant literature, we explored the narrative reflections of Black undergraduate women regarding their experiences of misogynoir and anti-racism with high school teachers.” (p. 2)
    • “Yet, less of this research considers how young Black women process their experiences with teachers after leaving their classrooms, or the extent to which it informs their motivation and sense of college readiness (Anderson & Martin, 2018). There is a need for critical scholarship on the academic experiences of high-achieving Black female learners to promote more nuanced representations of Black women’s achievement and Black girls’ educational trajectories (Anderson, 2020; Evans-Winters, 2014; Young, 2020). In the present qualitative study, we sought to explore how a sample of Black undergraduate women in the U.S. made sense of their high school journey, considering specifically the positive and negative experiences that participants described with teachers and school administrators. This reflective approach allowed us to engage the women in questions about how teachers helped or hindered them throughout high school…” (p. 2 & p. 6)
  • This was also included in the Limitations:
    • “First, we used a reflective qualitative approach with Black undergraduate women regarding their high school experiences, rather than talking with Black girls currently in high school. This was an intentional theoretical and methodological choice to capture the women’s significant memories about teachers after they had successfully graduated from the school; however, we might have gained different insights if we collected data with Black girls currently enrolled in high school…” (p. 21)

Methods 

The description of your participants who identified as African should be elaborated more. 22 identified as African and you state that they are "from" different countries. I assume that they all migrated to the US before starting high school (and that their entire high school education is from the US)? The age of migration/years in the US normally affect the way you see your experiences of racism and therefore it should be specified how long the 22 interviewees have been in the US. There are previous studies which show the different experiences of racism and exclusion that African American population and the immigrant African population face. This is why I believe that information on when they migrated and how long they have been in the country is relevant.

  • We appreciate this feedback, and we added a sentence about what we know of how long each woman spent in her parent’s country of origin (p. 7). To the reviewer’s other question, all of the women were U.S. citizens. We provided additional details in Table 1 on each woman’s ethnic identification, and we agree with the author about the potential divergence in African American and immigrant Africans’ understanding of race and experiences with racism.

There is no explanation of the racial demographics at either school and there should be. 

  • We added this to the Participants section in the Methods (i.e., “Twenty-one women attended a public institution in the Midwest (5% Black / African American) and the remaining 29 attended a selective, public institution in the Southeast (6% Black / African American) (p. 6).

The description of how this study was conducted and the process to secure inter-rater reliability among the authors is thoroughly detailed. I would have liked to see a table that described participants in the study including the neighborhood context of their schools (city, rural, or suburban), SES, IB/AP/honor roll, the college each participant was enrolled in and whether or not they were private and public, etc. 

  • We provided a demographic table with more information on participants (Table 1), including hometown and the women’s perceived high school racial composition (% of the student body that was Black). The only information that we have about their high school from the pre-interview survey is the women’s perceptions of their high school’s racial composition of the student body. As the reviewer mentioned, since we talked with college aged women about their K-12 schooling experiences rather than high school aged Black girls, we did not collect as much specific information about their schools and/or class enrollment (i.e., IB/AP/honor roll). We already included information on SES in the first version of the manuscript (i.e., self-perceived social class status and household income, p. 7). 

Design – should state the type of qualitative approach that was used – phenomenological, etc. or whatever they used to frame the design, data collection approach

  • As mentioned in the Coding Analysis Approach section, we used consensual qualitative research methods as our qualitative design (i.e., Consistent with recommendations from Hill (2012) on consensual qualitative research (CQR) methodology, the coding team (SL, PB, TP, and NW) used a consensual coding approach supervised by the lead author. CQR methods rely on the depth and richness of constructivist qualitative methods and reaching consensus among the coding research team on core ideas from the data (e.g., Ponterotto, 2005) (p. 9-10). 
  • We added more information about our data collection process (i.e., we collected individual interview data using a semi-structured interview protocol. The interview protocol included a set list of questions about race and gender socialization messages, as well as supplemental questions that interviewers could ask participants based on the direction and flow of the conversation (Cohen & Crabtree, 2006). This approach allowed the research team to collect reliable, comparative interview data on the topics of interest, while also providing the opportunity to identify new understandings through additional probing questions, p. 8-9).

Participants and Setting – the authors do not indicate any inclusion/exclusion criteria other than they attended one of the two PWIs and were African American or African. 

  • We added a statement on inclusion criteria (i.e., Students who identified as Black / African American women were eligible for participation in the study, p. 8)

A table of sociodemographic data would provide an overview of the population and would be useful.

  • We added a demographic table with participant information (Table 1, p. 7-8). 

Procedures – there is no mention of the IRB process and/or approval by the granting institution. It would be helpful to state how many individuals were on the research team at each institution and their roles.

  • We confirmed that we had IRB approval at each location(i.e., After receiving university IRB approval from each location, the PI sent weekly emails to Black student organizations until we scheduled the target number of participants for interviews, p. 8)
  • We already had a sentence about who was on the interviewing team (i.e., the interview team at each institution consisted of the PI (a Black woman, SL), as well as an additional Black woman graduate student (MW and DJ) per institution, who was trained in qualitative interview techniques.). We also added a new researcher positionality statement section (p. 9) to discuss the authors on the manuscript. 

Data Collection – The authors need to mention the year when the data was collected and the timeframe for data collection, including attrition.

  • We added information about the data collection (i.e., at the first institution, the PI had research funds to interview approximately 20 women. Twenty four women scheduled interviews (3 women canceled), and the 21 interviews were completed in the spring of 2019. At the second institution, the PI decided that theoretical saturation (Saunders et al., 2018) had been reached after 29 interviews. Thirty five women scheduled interviews (six women canceled), and the 29 interviews were completed in the fall of 2019), as well as general information about how many women scheduled interviews and canceled (p. 8; assuming this is what the reviewer meant by attrition, since this project was not longitudinal).

Interview protocol – The authors need to state how long the interviews were and what they used to record them. The authors could include tables that include topics, questions, etc. for the interviews.

  • We provided interview length (i.e., interviews lasted between 45-90 minutes (M = 75 minutes, p. 8). We added the method of recording (i.e., the interview team conducted individual interviews in safe, public locations that allowed for private conversations (i.e., reserved conference room and PI office) and used IPads to audio record the women’s narratives, p. 8) and we included an appendix with an abridged version of the interview protocol. 

Findings

I did not find the tables with the summary of discriminatory teacher practices and anti-racist teacher practices particularly useful because you provide in-depth analyses of these incidents in your findings. Your findings do not necessarily analytically engage misogynoir as an analytic. Instead, you provide descriptions of the different forms of discrimination participants went through. In regard to the anti-racist work, most of the data pulled shows Black girls engaging in this work rather than their teachers. The teachers were verbally supportive but lacked transformative action on behalf of their students.

  • We think the reviewer raises a valid critique and an important limitation of the current study - very few of the (White) teachers at least, contributed to transformative change from what we know from the women in our study. In the original submission, we mentioned that we had fewer examples of teachers engaging in anti-racism from the women’s narratives, which highlights the importance of integrating more interaction-level and institutional change in K-12 settings. In the Discussion, we tried to make a stronger case about this issue (p. 19) and we spoke with the editor about whether we should remove the anti-racist themes from the current manuscript. 
  • Based on recommendations from the editor, we kept the illustrative example tables in the manuscript (Tables 2 & 3). Based on other reviewer feedback, these tables provided quick glimpses at our findings, even though we do provide definitions and intensive descriptions in the main body of the text, as well. 
  • We also provided more information in the Introduction about misogynoir to further elucidate how many of our themes (i.e., body and tone policing) engage misogynoir as an analytic.

Discussion

The study aim/purpose should be included in the first paragraph.

  • We added the study aim/purpose to the first paragraph (p.18).
    • “The purpose of the current study was to explore how Black undergraduate women described the academic and social experiences they had with high school teachers, with a particular focus on misogynoir and anti-racism. In order to underscore the importance of dismantling racism and promoting racial justice in the classroom, the current reflective study provides a critical analysis of how Black college women experienced teacher support or marginalization during high school.”

While not requested, we reorganized the Discussion to mirror the Introduction: (1) recap of the main findings on misogynoir, (2) focusing on Black girls’ ways of knowing, and (3) discussing teacher practices in the classroom. We hope that this consistency helps with the flow of the narrative.

The discussion section provides the reader with the significance of this study and larger conversations necessary to transform the schooling experiences of Black girls. A lot of this information would strengthen the findings section in terms of the analysis and supporting claims made.

  • We appreciate this feedback. However, per editorial feedback, we did not pull from the Discussion to include cited evidence in the Findings section. However, we did flesh out the Implications section and include more institutional and structural level changes that could help transform the schooling experiences of Black girls.

The section on limitations is useful and could be expanded slightly. The inclusion of African young women without explanation is a limitation, especially since most of the literature review describes African American adolescent girls.

  • We added this to the Limitation section (p. 22).
    • “Also, a growing number of studies document the unique racialized experiences and perspectives among Black African immigrant students (e.g., Daoud et al., 2018; Hernandez, 2012; Mwangi & Fries-Britt, 2015). While all of the women in the current sample attended schools in the U.S., and thus, seemed to share certain types of racialized and gendered experiences with high school educators, it is important to call attention to within-group diversity in ethnically diverse Black students’ schooling experiences. If we had coded for qualitative differences in the misogynoiristic experiences of African American women and Black African immigrant women, the data might have revealed nuanced understandings of how Blackness is defined in America and Black African immigrants’ social positioning in U.S. classrooms (Mwangi & Fries-Britt, 2015). Researchers affirm that girls from Black African households receive unique socialization from parents and family members regarding education and sociological understandings of race (Kiramba et al., 2020), which may have informed their interactions with teachers and perceptions of their school settings. Still, much of this work looks at foreign-born Black immigrant students (for review, see Daoud et al., 2018) and the current sample included women who were born and raised in the U.S. Moreover, while the young women in the study may have possessed divergent beliefs about how race and gender operate, their teachers may have been less likely to know about the economic, social, and cultural distinctions among ethnically diverse Black girls in their classroom (Tatum, 2017). Thus, their interactions with the young women were likely informed by homogenizing understandings about Blackness and race.”

Further, if they were at HBCUs there may have been greater success in recruiting African American young women. This should be noted in future research.

  • We did not have trouble recruiting African American young women in the current study - at either PWI. We concluded our sampling at both institutions based on available grant funding (University 1) and theoretical saturation (University 2). While collecting data from Black women from diverse institutional contexts is an important area of future, we did not think it made sense to add this to the Limitations section since the women attended a range of high schools (including predominantly Black schools, as noted in Table 1). It would be interesting to know how Black girls make choices about their college attendance (opting to attend a PWI compared to an HBCU) and how that relates to their beliefs and perspectives on race, but that is well beyond the scope of the current study.

Limitations should be one section and expanded to describe how the study limitations could be addressed.

  • We modified the “Limitations and Future Research Directions” to “Limitations,” and made sure that we described how other researchers might use different conceptual framing or methods in future research. We also noted that other published manuscripts in Social Sciences includes a similar organization (e.g., Bichler et al. (2020) - Evolving patterns of aggression; Vanderlinden et al. (2020) - Motherhood in Europe).

This paper is rich in empirical material and has an inductive approach. In the concluding discussions of implications, the depth is lacking. One way to deepen the theoretical discussion can be to connect the empirical material and the concluding remarks with the concept of "oppositional gaze" more explicitly.

  • We think the author has valid feedback regarding the discussion. Based on this, we added the following: 
    • “Teacher preparation programs should be at the forefront of attending to critical discourse that challenges systemic oppression based on negative images and perceptions of Black girls. To combat misogynoir through oppositional gaze, teachers should do the following: 1) acknowledge and be willing to address their own biases, 2) develop tools that can assist with identifying and intervening when Black girls are being harassed or discriminated against, 3) work to understand concepts such as code-switching and behavioral norms from various backgrounds as not to punish what may be misinterpreted as misbehaving, 4) constantly evaluate data and outcomes of Black girls at the classroom and school-wide level to work towards collaborative change, and 5) be an advocate for Black girls who are learning to utilize oppositional gaze as a form of resistance. Instead of critiquing Black girls, these practices seek to understand their actions and intersectional identity through personal accountability and reflection.”  (p. 22-23)

Authors provide interactional level interventions to remedy discriminatory and misogynoir practices, but what are the institutional level changes that can be made to change practices in school?  

  • Thank you for this feedback. We changed the final paragraph in “Implications for Scholarly and Educational Praxis” to focus on institutional level changes. We also added a section on this in the Introduction, “Dismantling School Structures that Harm Black girls.”
    • “Finally, our findings are particularly important in thinking about the relevance of incorporating intersectional anti-racist practices into school-wide policies. This should involve inclusive recruitment and hiring practices for more educators of color; statistics show that the K-12 educational workforce (teachers and principals included) is still majority White (U.S. Department of Education, 2016). School administrators should provide professional development opportunities, consistent evaluation, and mentoring for teachers who are committed to anti-racism and equity. There are a number of scholarly and community experts who specialize in anti-racist leadership who can lend support to transform schooling contexts (Superville, 2020). In addition, school leaders must be open to addressing racist incidents that occur inside and outside of the school with students and model how to engage in proactive dialogue around race, colorblindness, and anti-racism. Educators and school leaders, particularly those who are White, must not assume that they know the best ways to support Black students; instead, they should be open to seeking and incorporating feedback from parents and students about ways to improve the school racial climate and classroom dynamics (Schniedewind & Tanis, 2017). Finally, anti-racist schooling systems do not occur in a vacuum – teachers and principals need support from superintendents and school boards to enact district-level change (Irby et al., 2019). Anti-racist work is a long-term, but critical commitment if educators want to create learning contexts that offer students equitable opportunities to thrive.” (p. 23-24)

Reviewer 4 Report

Hats off to the authors for such an essential and needed focus for Black girls and their educational experiences! The author(s) address a topic and population that is understudied, timely and long overdue for education and anti-racist practice. The study is based on data collected from Black female college students to understand their experiences with misogyny and racism when they were in high school. Overall, the paper is well written and not riddled with grammatical errors. However, enthusiasm for the study is slightly dampened given the need for focus in the intro/background/literature review, theoretical conceptualization and editing and revising of the paper that reflects the overall goal of the manuscript. The authors developed a clear description of the plight of young Black females. There are some significant limitations to the paper, as I describe below. These and other specific concerns are detailed below with recommendations for how the manuscript might be improved.

Introduction/Literature Review

  1. Pages 2-4 primarily focus on adolescent girls and should actually focus on both adolescent and young adult/college age women. This will provide a more balanced perspective about the study participants. Also, some of this content could be cut down especially the focus on girls being arrested in high school since that was not the focus or noted in the findings for the current study.
  2. The authors include three sections in the background about anti-racist praxis, politicized ethic of care and restorative educational justice that are not clearly defined/explained as perspectives and/or content to inform the study and design, etc. One suggestion would be to add an overview – a few sentences before the paragraph about anti-racist praxis to explain the purpose of this information and why it’s needed.

Methods

  1. Design – should state the type of qualitative approach that was used – phenomenological, etc. or whatever they used to frame the design, data collection approach
  2. Participants and Setting – the authors do not indicate any inclusion/exclusion criteria other than they attended one of the two PWIs and were African American or African. Further, nearly half of the sample included African young women so their experiences are likely different than those of African American young women but this is not explained in the results, discussion – it should be noted and explained or note the lack of this distinction in the limitations.
    1. A table of sociodemographic data would provide an overview of the population would be useful.
  3. Procedures – there is no mention of the IRB process and/or approval by the granting institution. It would be helpful to state how many individuals were on the research team at each institution and their roles.
  4. Data Collection – The authors need to mention the year when the data was collected and the timeframe for data collection, including attrition.
  5. interview protocol – The authors need to state how long the interviews were and what they used to record them. The authors could include tables that include topics, questions, etc. for the interviews.

Results

The authors have several pages of study results, which could have been parsed out if there were a theory guiding the study, i.e, critical race theory or socio-ecological theory. However, reporting them based on the three areas of content from the background section of the paper was prudent.

Discussion

First, it would be useful to include a summary of major findings to highlight the main themes of the study. Other issues are as follows:

  1. The study aim/purpose should be included in the first paragraph.
  2. The section on limitations is useful could be expanded slightly. The inclusion of African young women without explanation is a limitation, especially since most of the literature review describes African American adolescent girls. It could be that since the studies were conducted at PWIs the population of African American females is much smaller and there is no explanation of the racial demographics at either school and there should be. Further, if they were at HBCUs there may have been greater success in more recruiting African American young women. This should be noted in future research.
  3. There isn’t an overarching theory to focus the study design, sample and analysis.
  4. Future research should all be in one section and expanded to describe how the study limitations could be addressed.

Author Response

(The authors gave the same response as above.)

Round 2

Reviewer 3 Report

Authors did a great job making revisions in this paper. 

Author Response

We appreciate the reviewer feedback and support on this manuscript, and attended to additional semantic edits, as requested by the editor.

Reviewer 4 Report

The authors addressed the reviewer's concerns conclusively. The paper should be accepted pending any additional editing and polishing.

Author Response

(The authors gave the same response as above.)
